# Systemic and Local Strategies for Primary Prevention of Breast Cancer

**DOI:** 10.3390/cancers16020248

**Published:** 2024-01-05

**Authors:** Erin K. Zaluzec, Lorenzo F. Sempere

**Affiliations:** 1Precision Health Program, Michigan State University, East Lansing, MI 48824, USA; zaluzece@msu.edu; 2Department of Pharmacology & Toxicology, College of Veterinary Medicine, Michigan State University, East Lansing, MI 48824, USA; 3Department of Radiology, College of Human Medicine, Michigan State University, East Lansing, MI 48824, USA

**Keywords:** breast cancer prevention, mammary gland, intraductal delivery, ductal tree, epithelial cell carcinogenesis, chemoprevention, endocrine therapy, transdermal gel

## Abstract

**Simple Summary:**

Current interventions for breast–cancer prevention are associated with adverse side effects that frequently deter women from selecting these evidence-based risk-reducing procedures. In addition, modifiable lifestyle changes can improve breast cancer risk but can be challenging in execution. Therefore, women who are considered high risk due to non-modifiable factors, such as *BRCA1/2* mutations or a strong family history of breast cancer, are in need of alternative prevention procedures. Here, we review investigational preclinical and clinical approaches at a systemic and local level that focus on non-modifiable breast cancer risk-reducing interventions.

**Abstract:**

One in eight women will develop breast cancer in the US. For women with moderate (15–20%) to average (12.5%) risk of breast cancer, there are few options available for risk reduction. For high-risk (>20%) women, such as *BRCA* mutation carriers, primary prevention strategies are limited to evidence-based surgical removal of breasts and/or ovaries and anti-estrogen treatment. Despite their effectiveness in risk reduction, not many high-risk individuals opt for surgical or hormonal interventions due to severe side effects and potentially life-changing outcomes as key deterrents. Thus, better communication about the benefits of existing strategies and the development of new strategies with minimal side effects are needed to offer women adequate risk-reducing interventions. We extensively review and discuss innovative investigational strategies for primary prevention. Most of these investigational strategies are at the pre-clinical stage, but some are already being evaluated in clinical trials and others are expected to lead to first-in-human clinical trials within 5 years. Likely, these strategies would be initially tested in high-risk individuals but may be applicable to lower-risk women, if shown to decrease risk at a similar rate to existing strategies, but with minimal side effects.

## 1. Introduction

For women in the US, breast cancer (BC) is the most prevalent cancer and the second-leading cause of cancer-related deaths. Projections for 2023 estimate that 55,720 women will be diagnosed with carcinoma in situ, 297,790 with invasive carcinoma, and 43,170 women will die from BC [1]. With the number of new diagnoses still on the rise, one in eight women will develop BC within their lifetime, but all women are at risk. Therefore, there is a need to develop new strategies for primary prevention with a focus on high-risk (>20%) individuals, but that can also be applied to moderate- (15–20%) and average (12.5%)-risk individuals.

There are well-established risk factors that contribute to the absolute risk of developing BC [2,3]. There are modifiable risk factors such as having a healthy diet, regular exercise, and limiting alcohol consumption [2,4]. Cumulative exposure of breast tissue to estrogen is an important risk factor for BC. This risk can be minimized by various actions such as having their first-born before age 30, limiting the use of hormonal birth control medications, and avoiding hormone replacement therapy. Non-modifiable risk factors that increase cumulative exposure to estrogen include younger age at menarche and older age at natural menopause [2]. There are non-modifiable genetic risk factors that include known mutations in a high-penetrant BC gene such as *BRCA1* and *BRCA2*, cumulative interaction of risk-associated alleles of BC susceptible SNPs, and/or family history with multiple incidences of BC [5]. There are also non-modifiable risk factors related to the personal history of radiation therapy to the chest, and the number of breast biopsies [4].

For women in the US, a 1.67% increased risk over 5 years at any age or 20% increased risk over a 20-year period is considered a high-risk individual [3]. This review is focused on federal agency guidelines and regulatory approvals that pertain specifically to US women. It is worth noting that some of these guidelines are different in other countries. For example, for women in Europe, high risk is defined as a >30% lifetime risk [6]. This difference and other considerations may affect how risk-reducing interventions are applied, perceived, and complied with in different geographical regions and countries. Many non-modifiable factors contribute to this increased risk. *BRCA* mutation carriers are considered high-risk individuals (>50% chance of BC development) and are the most prevalent and counseled group for primary intervention. Women with other genetic predispositions including mutation in *CDH1* (hereditary diffuse gastric cancer syndrome), *PALB2, PTEN* (Bannayan–Riley–Ruvalcaba, Cowden, or hamartoma tumor syndromes), *TP53* (Li-Fraumeni syndrome), *STK11* (Peutz–Jeghers syndrome) are also considered high-risk individuals and eligible for bilateral prophylactic mastectomy [7,8].

Guidelines for risk reduction of moderate (15–20%)-risk individuals are not as well-delineated in part due to the difficulty of identifying this subset of individuals and how modifiable risk factors such as diet, alcohol consumption, and BMI can affect and compound absolute risk [4,5]. However, women with genetic predispositions including mutations in *ATM* and *CHK2,* or who carry risk-associated alleles for multiple of the 92 susceptibility genes are generally considered moderate-risk individuals [9]. Genetic counseling based on multi-gene panels captures the most frequent mutations [7,8,10,11]. Recommendation for mammography and other monitoring modalities varies by risk [3,12].

There are several risk assessment models available that calculate an individual’s risk based on various risk factors [2,13]. The BCRAT tool (https://bcrisktool.cancer.gov/ (accessed on 30 December 2023)) uses the Gail model and is appropriate for women without a genetic predisposition or previous BC [14]. The IBIS tool (https://ibis.ikonopedia.com/ (accessed on 30 December 2023)) uses the Tyrer–Cuzick model and is appropriate for women with known or suspected genetic predisposition, including mutations in *BRCA1* or *BRCA2*, using extensive personal and family history risk factors [15].

For high-risk women in the US, FDA-approved primary prevention strategies include surgical removal of the breasts and/or ovaries and the use of anti-estrogen therapies. Bilateral prophylactic mastectomy is currently the most effective procedure for preventing BC: it can reduce the incidence of BC by up to 90% in high-risk individuals [16]. Anti-estrogen treatments have been shown to reduce BC risk by up to 50% in high-risk women [16]. Though effective in risk reduction, less than 50% and less than 10% of high-risk individuals opt for surgical or hormonal interventions, respectively, with life-changing consequences and severe side effects as major contributing dissuading factors [16,17,18]. These prevention interventions are readily available but may be underused due to a lack of clinician or patient information regarding risk level, lack of clinical confidence to discuss appropriate prevention options, personal social dynamics, and fully informed choice, which can result in a low uptake of prevention methods [16]. Bilateral prophylactic mastectomy uptake is well documented in women who are *BRCA* mutation carriers. However, on average, only 20% of women at high risk without the *BRCA* mutations undergo this surgical procedure but have reported ranges from 11–50% [19,20]. Population studies on hormonal interventions have reported low uptake for eligible women (1–5%); however, this falls short when compared to high-risk proactive women interested in using these interventions, which can be as high as 40% [16,21]. A recent study conducted in Europe showed that women who were informed of their risk and provided information on preventive options within 8 weeks of risk identification had a large increase in uptake (77.5%) of hormonal interventions compared to much lower uptake in standard clinical settings (11–20)% [18,22,23,24]. Therefore, prevention interventions, either currently approved or investigational, should take into consideration education and informed decision-making in addition to clinically established risk reduction and management of adverse side effects.

These primary prevention interventions treat the entire breast as a unit, but neoplasia originates in discrete regions within one or more ductal tree systems [25,26]. A woman’s breast contains the stroma and 8–12 ductal trees [27,28,29]. Surrounding the ductal tree is the stroma, which is composed of adipocytes, immune cells, endothelial cells, fibroblasts, and extracellular matrix (Figure 1A). The ductal trees are composed of luminal epithelial cells, myoepithelial cells, and mammary epithelial stem cells from which most BC arises and are not readily accessible without highly invasive surgical procedures. This review focuses on preclinical and clinical approaches that target the premalignant epithelial niche and minimize adverse side effects while maintaining the effectiveness of current surgical and systemic interventions for women with non-modifiable risk.

## 2. Evidence-Based Interventions for Primary Prevention

We briefly review the limited number of FDA-approved drugs and interventions for primary prevention of BC. Due to adverse side effects, recommendation for these interventions is generally restricted to high-risk individuals. 

### 2.1. Surgical Intervention

Surgical interventions target areas of the breast that are associated with a higher risk of BC development through the complete removal of the tissue. Due to the invasive nature of these procedures, only high-risk women consider these options. Dual mastectomy directly removes both breasts and in doing so removes the ductal tree, whereas salpingo-oophorectomy removes fallopian tubes and/or ovaries for reduced hormonal contribution in BC development. Here, we discuss the advantages and disadvantages of surgical intervention.

*Bilateral prophylactic mastectomy.* This procedure completely removes the breast tissue in both breasts, including the ductal tree and surrounding stroma. Bilateral prophylactic mastectomy is a highly invasive procedure that can be executed as a total mastectomy, skin-sparing mastectomy, or total skin-sparing mastectomy. In removing the entirety of the breast, this risk-reducing surgery removes the epithelial cells, the intended target cells, from which breast carcinomas arise [30,31]. With nipple reconstructive techniques available, total mastectomy is the most commonly selected choice [32]. However, despite risk reduction and available reconstructive surgery, less than 50% of women undergo this preventative procedure (Figure 1B) due to the pain, cosmetic, psychological, and social impact [17,33,34]. Pain, local, and systemic inflammation in response to mastectomy varies on the surgical procedure which can include musculoskeletal manipulation and tissue advancement for reconstruction in addition to the excision of the breast tissue. Bilateral prophylactic mastectomy studies focusing on patient-reported outcomes after breast reconstruction have noted higher body pain or breast discomfort and decreased sexual interest, but overall satisfaction with the procedure lowered cancer-related anxiety and increased satisfaction with breast cosmesis [35,36,37,38,39,40]. However, it is important to note that these reports vary due to factors such as number of patients, the timing of reconstructive surgery, and pre- vs. post-operation comparisons [35]. Nevertheless, the potential positive and negative impacts should be equally addressed with women for them to make fully informed decisions.

*Salpingo-oophorectomy.* Surgical removal of the fallopian tubes and ovaries is a commonly recommended BC prevention for premenopausal *BRCA* mutation carriers that had a previously reported risk reduction of up to 50% [41]. These findings have been called into question after considering the salpingo-oophorectomy procedure as a time-varying covariate. Recent studies investigating BC risk reduction in *BRCA* mutation carriers following salpingo-oophorectomy have found minimal to no impact in women when taking into consideration the time-varying covariate [41,42,43]. Following salpingo-oophorectomy, women may experience symptoms typically associated with menopause due to the reduction of estrogen and progesterone production in the body. However, these symptoms are often reportedly tolerable and do not require further treatment [44]. Hormone replacement therapy for women experiencing severe symptoms is available. *BRCA* mutation carriers taking hormone replacement therapy after undergoing oophorectomy do not have an increased risk of BC if <45 years old but those >45 have an increased risk of triple-negative BC [45]. Other studies have shown that *BRCA* mutation carriers who receive only estrogen after an oophorectomy have no increased risk of BC. However, the risk of only progesterone use has yet to be determined [46,47]. 

### 2.2. Hormonal Modulation

Selective estrogen receptor modulators (SERMs), tamoxifen and raloxifene, are commonly used in the treatment of various diseases including BC and osteoporosis, and are the only two FDA-approved compounds for primary prevention of BC [5,48]. An estrogen-bound estrogen receptor (ER) forms a complex that hetero- or homo-dimerizes with a second estrogen-bound estrogen receptor. This complex is able to translocate into the nucleus and can act directly and indirectly on genes to promote cell growth, migration, and metastasis while simultaneously preventing functions such as apoptosis and necrosis [49,50,51]. These SERMs compete with and block the binding of estrogen to the ER within the breast epithelia and antagonize its functions (Figure 1C and Figure 2A); interaction of these SERMs with ERs in other cell types throughout the body can have partial antagonistic or even agonistic effects [52].

*Tamoxifen.* Tamoxifen was originally developed in the 1960s as a contraceptive but was ultimately unsuccessful. In the 1990s, it was repurposed as a BC treatment due to its therapeutic benefits in reducing tumor recurrence at the origin site, lowering the incidence of cancer in the contralateral breast, and increasing overall survival. These benefits led to the selection of tamoxifen as a preventive agent for BC [53,54]. By 1992, the National Surgical Adjuvant Breast and Bowel Project had enrolled 13,175 high-risk women in a trial investigating the role of tamoxifen as an oral drug for the prevention of primary formation of BC. By the end of the five-year clinical trial, tamoxifen had reduced the incidence of invasive and non-invasive BC by 49% and 50%, respectively, compared to placebo groups [53]. Women who developed invasive BC had lower occurrence and tumor size in the tamoxifen-treated groups. Despite the reduction of BC incidence, increased hot flashes and vaginal discharge were reported among women in the tamoxifen-treated groups. More alarming was the associated risk of stroke and a 1.5-to-6.9-fold increased risk of developing endometrial cancer due to long-term exposure to tamoxifen [53,55,56]. Additional studies of tamoxifen as a preventive agent reported these adverse effects as major factors contributing to discontinuation of treatment, especially among women taking tamoxifen for primary prevention as opposed to adjuvant therapy for BC [57,58]. However, adverse effects ceased after treatment, and long-term follow-up studies showed extended tamoxifen protection for BC prevention [59,60,61]. 

*Raloxifene.* Similar to tamoxifen, raloxifene was originally developed for non-BC disease. Initially under investigation for treatment and prevention of osteoporosis in postmenopausal women, the Multiple Outcomes of Raloxifene Evaluation (MORE) study in 1998 was conducted to investigate bone fractures in 7700 post-menopausal women diagnosed with osteoporosis [62]. Interestingly, during this study, raloxifene was associated with a lower incidence of BC. Raloxifene treatment resulted in a 76% decrease in invasive BC after 3 years of treatment [62]. Like tamoxifen, reports of hot flashes along with leg cramps were higher among women taking raloxifene. Additionally, thromboembolic events were 3.1 times higher with raloxifene but did not increase the risk of endometrial cancer [62,63]. Direct comparison of tamoxifen and raloxifene revealed equal risk reduction of invasive BC, but endometrial cancer in postmenopausal women, thromboembolic events, and stroke occurred in both groups [64]. The raloxifene-treated group had a 30% lower rate of endometrial cancer, had fewer side effects on the uterus, and had a lower incidence of thromboembolic events which was consistent over the 81-month follow-up [64,65]. 

### 2.3. Watchful Waiting 

Despite the evidence-based effectiveness of these surgical procedures and hormonal interventions, up to 50–70% of women presented with prevention and treatment options choose a watchful waiting strategy with enhanced surveillance [66,67]. These strategies may include annual or more frequent mammography and magnetic resonance imaging, monthly self-examination, and other monitoring protocols. Watchful waiting strategies do not reduce the risk of developing BC (Figure 1A) and up to 70% of high-risk individuals do not adhere to their enhanced surveillance protocols decreasing the benefit of early detection and intervention [68,69,70]. 

## 3. Investigational Approaches for Primary Prevention

There are several investigational approaches that aim to reduce or limit adverse side effects of current interventions and/or develop novel interventions with a high safety profile that may be offered more broadly to women seeking proactive options for primary prevention of BC. These approaches can be divided into systemic and local approaches. Systemic approaches consist of intravenous, intramuscular, or oral delivery of a drug, whereas local approaches consist of intraductal (ID) injection, transdermal application, or subcutaneous implant for drug delivery (Figure 1D,E, Figure 2B–D and Figure 3B–E). Some of these approaches have already reached the clinical trial stage (Table 1) as discussed below, whereas many are still at an early stage of development in preclinical animal models (Table 2 and Table 3).

**Table 1 cancers-16-00248-t001:** New approaches for primary prevention of breast cancer in clinical trials.

Approach	Intervention	Active Agent	Number of Participants	Results	Reference(s)
Hormonal therapy	Local	Endoxifen gel	90	1.9% Reduction in mammographic density	NCT04616430, Completed [71]
		4-Hydroxytamoxifen transdermal gel	194	52% Decrease in Ki-67 labeling index	NCT03063619, Active [72]
		Fulvestrant	3	N/A	NCT02540330, Terminated
	Systemic	Aromatase inhibitors (Anastrozole)	3864	N/A	NCT00078832, Completed
		Aromatase inhibitors (Letrozole)	55	N/A	NCT00579826 Completed
Chemoprevention		Retinoid (Fenretinide)	20	≤50% Risk reduction	NCT01479192 [58,73] Terminated

Preclinical models of primary prevention and local treatment of BC include genetically engineered mouse models, chemical carcinogen-induced rat models, and orthotopic cell line models. These models present different advantages and limitations. Genetically engineered mouse models (GEMMs) have, in principle, an inexhaustible potential of developing malignancy, whereas chemical carcinogen-induced and orthotopic models have a limited number of neoplastic cells. In some of these models, the line between primary prevention and local treatment is difficult to establish. Different GEMMs have been used to model human BC (Table 2 and Table 3). Several studies have used similar *Brca1*-deficient GEMMs as they are more relevant to high-risk *BRCA* mutation carriers. However, other GEMMs, including MMTV-PyMT, MMTV-Erbb2, and C3(1)-TAg, often considered more aggressive models, have also been used in this setting. Chemically induced rat models have been used for almost 40 years to study hormonal control [74] and more recently have been used to investigate local approaches for BC prevention (Table 3). Common limitations of these rodent models include the relative simplicity of linear architectures of their ductal trees, smaller ductal tree volumes with different surface area to volume ratios, different stromal density compared to human counterparts for translation of local intervention, duration, and frequency of chronic treatment in rodents vs. humans, and establishing effective dose for translation of systemic interventions.

**Table 2 cancers-16-00248-t002:** Systemic approaches for breast cancer prevention in preclinical models.

Approach	Active Agent	Experimental Model ^1^	Level of Evidence ^2^	Results	Reference(s)
Prophylactic Vaccine	α-Lactalbumin	MMTV-HER2 (*n* = 6)	A	Increased latency (*p*-value = 0.0004)	[75]
		MMTV-PyMT (*n* = 8)	D	Reduced tumor burden (*p*-value < 0.0006)	
		4T1 isograft (*n* = 8)	D	Reduced tumor burden until day 13 post injection (*p* value = 0.0006)	
	HER2	MMTV-HER2 (*n* = 10)	A	Increased latency (*p*-value < 0.01)	[76]
	HER2	MMTV-HER2 (*n* = 5–8)	A	Increased latency (*p*-value < 0.02)	[77]
Chemoprevention	Erlotinib	*Brca1*^fl/fl^;Trp53^+/−^; MMTV-Cre (*n* = 13)	A	Increased latency (*p*-value = 0.0001)	[78]
	I-BET 762	MMTV-PyMT (*n* = 13)	B	Increased latency (*p*-value < 0.05)	[79]
	CCDO-Me	*Brca1*^fl/fl^;Tpr53^+/^; MMTV-Cre (*n* = 15)	A	Increased latency (*p*-value < 0.05)	[80]
	RankL inhibitor	*Brca1*^fl/fl^;Trp53^+/−^; MMTV-Cre (*n* = 17)	A	Increased latency (*p*-value < 0.001)	[81]
	RankL monoclonal antibody	*Brca1*^fl/fl^; MMTV-Cre (*n* = 9)	A	Increased latency (*p*-value < 0.001)	[82]
	Cox-2 inhibitor	MMTV-Erbb2 (*n* = 24)	A	Reduced tumor incidence (*p*-value = 0.003)	[83]
	Curcumin	4T1 isograft (*n* = 9)	C	Reduced tumor burden (*p*-value < 0.05)	[84]
	Bisphosphonates (zolendronic acid and risdronate)	MDA-MB-231 xenograft (*n* = 12)	D	Reduced tumor burden (*p*-value < 0.05)	[85]
	Rexinoids (Bexarotene)	MMTV-Erbb2 (*n* = 20)	A	Increased latency (*p*-value < 0.0001)	[86]
		MMTV-Erbb2 (*n* = 19)	A	Increased latency (*p*-value < 0.001)	[87]
	JAK3 and EGFR inhibitor (WHI-P131	DMBA-induced Balb/c mice (*n* = 20)	B	Increased latency (*p*-value = 0.0014)	[88]
Cytotoxic	Paclitaxel	DMBA-induced Balb/c mice (*n* = 20)	B	Increased latency (*p*-value = 0.0041)	[88]

*Notes*: ^1^ “*n*” denotes the number of animals in investigational treatment group instead of overall number of animals in all groups of the study. ^2^ A = animals were observed for 6 months to 2 years; B = for 8 weeks to 6 months; C = for 4 to 8 weeks; D = for <4 weeks. *Abbreviations:* Brca: breast cancer gene, CDDO-me: 2-Cyano-3,12-dioxooleana-1,9(11)-dien-28-oic acid methyl ester, Cox-2: Cyclooxygenase-2, Cre: cre-recombinase, DMBA: 4 7,12-Dimethylbenz[a]anthracene, EGFR: epidermal growth factor receptor, Erbb2: avian erythroblastic leukemia viral oncogene homolog 2, HER2: human epidermal growth factor receptor 2, I-Bet: inhibitor of Bromodomain and extra-terminal domain, Jak3: Janus Kinase 3, MMTV: Mouse Mammary Tumor Virus, Py-MT: polyoma middle tumor-antigen, RankL: receptor activator of nuclear factor kappa beta ligand, Trp53: Tumor Protein P53.

**Table 3 cancers-16-00248-t003:** Intraductal and local approaches for breast cancer prevention in preclinical models.

Intervention	Active Agent	Experimental Model ^1^	Level of Evidence ^2^	Results	Reference(s)
Chemoprevention	Oral-free curcumin	MNU-induced Sprague Dawley rats (*n* = 12)	A	Reduced tumor incidence (HR = 3.95, *p*-value 0.007)	[89]
	Intraductal free curcumin			Reduced tumor incidence (HR = 2.85, *p*-value 0.020)	
	Nanocurc encapsulated curcumin		A	Reduced tumor incidence (HR = 2.88, *p*-value 0.028)	
Cytotoxic	Paclitaxel	MNU-induced Sprague-Dawley rats (*n* = 15)	A	Reduced tumor burden (*p*-value < 0.05)	[90]
	Pegylated Liposomal Doxorubicin	MMTV-Erbb2 (*n* = 12)	B	Reduced tumor incidence (HR = 6.40, *p*-value < 0.0001)	[91]
		MNU-induced Sprague Dawley rats (*n* = 15)	B	Reduced tumor incidence (*p*-value < 0.001)	[91]
		MNU-induced Sprague Dawley rats (*n* = 5)	B	No change compared to control	[92]
	5-fluorouracil	MNU-induced Sprague Dawley rats (*n* = 5)	B	Reduced tumor incidence (HR = 3.30, *p*-value = 0.018)	
	Carboplatin	MNU-induced Sprague Dawley rats (*n* = 5)	B	Reduced tumor incidence (HR = 10.4, *p*-value < 0.0001)	
	Nanoparticle albumin-bound paclitaxel	MNU-induced Sprague Dawley rats (*n* = 5)	B	No change compared to control	
	Methotrexate	MNU-induced Sprague Dawley rats (*n* = 5)	B	No change compared to control	
	Nanoparticle albumin-bound paclitaxel	MNU-induced Sprague Dawley rats (*n* = 6)	B	Reduced tumor burden (*p*-value < 0.05)	[93]
	Cisplatin	*Brca1*^fl/fl^Trp53^L/L^; WAPcre (*n* = 20)	A	Increased latency (*p*-value < 0.0001)	[94]
Hormonal therapy	4-hydroxytamoxifen (4-OHT)	MNU-induced Sprague Dawley rats (*n* = 20)	A	Reduce tumor incidence (*p*-value < 0.0001)	[91]
	Fulvestrant	MIND MCF-7 xenograft (*n* = 3)	B	Reduced tumor burden (*p*-value < 0.001)	[95]
		MNU-induced Sprague Dawley rats (*n* = 10)	B	Increased latency (*p* < 0.0001), reduced tumor incidence (HR = 2.08)	
	Fulvestrant and silastic tubing	MCF-7 xenograft (*n* = 8)	C	Reduced tumor burden (*p*-value < 0.05)	[96]
Suicidal gene vector	Adenovirus vector with thymidine kinase and gancyclovir	MNU-induced Wistar Furth rats (*n* = 30)	A	(paradoxical) Decreased latency and increased tumor incidence	[97]
Gene silencing	Liposomal Hox1A siRNA silencing	C3(1)-TAg (*n* = 8)	B	Reduced tumor Incidence	[98]
Radioimmunotherapy	Radio-conjugated trastuzumab	MIND SUM225 xenograft (*n* = 3, 4)	C	Dose-dependent reduced tumor burden	[99]
Targeted immunotoxin	Anti-transferrin receptor-antibody conjugated pseudomonas exotoxin	MIND MCF7 xenograft (*n* = 20)	C	Increased latency and reduced tumor burden (*p*-value < 0.001)	[100]
Chemical ablation	Ethanol	C3(1)-TAg (*n* = 13)	A	Increased latency (*p*-value < 0.0001), reduced incidence (HR = 4.76, *p*-value < 0.0001)	[101]

*Notes*: ^1^ “*n*” denotes the number of animals in the investigational treatment group instead of the overall number of animals in all groups of the study. ^2^ A = animals were observed for 6 months to 2 years; B = for 8 weeks to 6 months; C = for 4 to 8 weeks; D = for <4 weeks. *Abbreviations:* Brca: breast cancer gene, Cre: cre-recombinase, Erbb2: avian erythroblastic leukemia viral oncogene homolog 2, Hox1A: homeobox protein 1A, MIND: Mouse Mammary Intraductal, MMTV: Mouse Mammary Tumor Virus, MNU: *N-methyl-N-nitrosourea*, siRNA: short interfering RNA, Trp53: Tumor Protein P53, WAPcre: Whey Acidic Protein Cre-Recombinase.

## 4. Systemic Approaches for Primary Prevention in Preclinical Models and Clinical Trials

Systemic interventions require ADME (absorption, distribution, metabolism, and excretion) processing of the drug leading to drug delivery throughout the body. There are many well-characterized drugs with known modes of action and systemic effects on the body. Scientists are now investigating the repurposing of some of these drugs for the primary prevention of BC.

### 4.1. Hormonal Therapy with Aromatase Inhibitors

Similar to tamoxifen and raloxifene, aromatase inhibitors are already approved for BC treatment. These drugs mimic androstenedione and bind to aromatase enzymes to prevent the conversion of testosterone to estrogens [102]. Type I inhibitors bind irreversibly to aromatase, whereas type II inhibitors have reversible, competitive inhibition [102]. Reduction in contralateral BC and overall survival improvement in the treatment of early-stage BC led researchers to investigate aromatase inhibitors for primary prevention. Currently, a type I inhibitor (exemestane) and two type II inhibitors (letrozole and anastrozole) are in clinical trials as preventive agents for BC. Exemestane clinical trials recruited 4500 high-risk, post-menopausal women for a median 3-year study [103]. Like tamoxifen, exemestane was taken orally and daily. Compared to the placebo, the exemestane group reported a 65% reduction in invasive BC incidence [103]. Hot flashes, fatigue, sweating, and insomnia were reported in both groups and mild bone density loss occurred in the exemestane group [104]. After a 5-year follow-up, no serious adverse effects, such as bone fractures, endometrial cancer, or thrombotic effects had occurred [103]. A phase 2 clinical trial (NCT00579826) with letrozole was recently completed and findings of this study have not yet been published.

The phase 3 clinical trial IBIS II (NCT00078832) with anastrozole involved 3864 post-menopausal women for a 10-year follow-up study [105]. Anastrozole was given orally and daily for 5 years. Compared to the placebo, the anastrozole group reported a 49% reduction in BC significantly within and after the first 5 years of treatment. Additionally, a significant decrease in non-BC was also observed. Arthralgia, joint stiffness, hot flashes, sweating, hypertension, and vulvovaginal dryness had higher reports among anastrozole groups compared to placebo treatment groups. Additionally, major adverse effects such as fractures, myocardial infarction, deep vein thrombosis, pulmonary embolism, stroke, and transient ischemic attacks were non-significantly different between anastrozole and placebo treatment groups [105]. Recently, anastrozole has received regulatory approval for BC prevention use in the United Kingdom; however, the FDA has yet to approve this or any other IA in a primary prevention setting [3]. 

### 4.2. Chemoprevention

Chemoprevention involves a variety of drugs used to prevent or delay the onset of cancer. These drugs are used as an alternative to invasive surgical procedures, such as bilateral prophylactic mastectomy, to lower a person’s risk of cancer development. Although chemoprevention drugs may have life-saving benefits, women can be hindered by the associated side effects. Therefore, most women who qualify for BC chemoprevention are high-risk individuals.

*Retinoids and Rexinoids.* Retinoids are vitamin A analogs that primarily bind to retinoid acid receptors for growth regulation, differentiation, and apoptosis. However, these drugs have limited use due to their toxicity. Fenretinide is a synthetic derivative of all-trans-retinoic acid that has been shown to accumulate in human breast tissue and prevent BC development in animal models [106,107,108] and has low toxicity compared to other retinoids. This led to a clinical trial for secondary prevention in pre- and post-menopausal women in 1987. A total of 1750 women participated in this 5-year study with up to a 15-year follow-up. Interestingly, pre-menopausal women had up to a 50% risk reduction of both ipsilateral and contralateral BC which was not seen in women over the age of 55 years. However, no significant difference was seen with distant metastases formation; new, non-breast, primary tumor formations; or overall mortality [58]. A follow-up study on the safety of fenretinide was conducted in 2800 women with stage I or ductal carcinoma in situ given the same dosage for 5 years. Under 20% of women reported diminished dark adaptations or dermatological effects, 13% reported gastrointestinal symptoms, which all decreased over the 5-year period, whereas 11% reported ocular surface disorders which had a slight increase in occurrence [73]. Liver function and lipid profile were also tested with no significant difference between the treatment and control groups. No significant differences were found between pre- and post-menopausal women for adverse effects [73]. These results led to a clinical trial for high-risk pre-menopausal women but it was terminated due to low patient accrual (NCT01479192). 

Retinoid X receptor is another class of retinoid receptors that has different affinities for naturally occurring ligands and can dimerize with a plethora of receptors such as vitamin D, thyroid hormone, and liver X receptors. This vast access to regulator networks makes this an ideal target for cancer prevention and treatment [109]. Currently, bexarotene, otherwise known as LGD1069, is a rexinoid that is selective for the retinoid X receptor. It is the only FDA-approved rexinoid for the treatment of cutaneous T-cell lymphoma. However, preclinical studies of bexarotene for BC prevention have led to active, ongoing clinical trials (NCT03323658, NCT00055991) in patients at high risk for BC development [86,87]. 

*Erlotinib.* Women with *BRCA* mutations do not frequently express hormonal receptors and develop triple-negative BC. However, epidermal growth factor receptor (EGFR) is highly expressed in these triple-negative BCs and is suggested to be associated with *BRCA1* mutations in this cancer subtype [110,111,112]. Erlotinib is a small molecule inhibitor that blocks the phosphorylation and activation of EGFR and has been tested as a chemopreventive agent in *Brca1*-deficient GEMM [78]. At three months of age, animals were given an oral daily dose of 100 mg/kg of erlotinib, equivalent to ~8 mg/kg in humans, to study tumor-free survival and tolerability of chronic erlotinib treatment. Alopecia was the main side effect observed and was only found in 30% of all animals [78]. Erlotinib treatment significantly delayed or prevented the formation of ER- tumors but had a minimal effect on the formation of ER+ tumors [78]. Implicit from these findings is that a combination treatment with hormone therapy and erlotinib could prevent the formation of both ER-dependent and ER-independent EGFR-dependent tumors. However, the combination treatment may also exacerbate the side effects of either drug.

*Bromodomain inhibitors.* Epigenetics acts on gene regulation that can impact cancer initiation and progression. Unlike genetic modifications, these events are considered reversible making them a desirable drug target. Several drugs have already been developed to modulate DNA methylation or histone modifications. One of the drug targets are chromatin readers such as the bromodomain and extra-terminal (BET) protein family whose conserved bromodomain binds to specific epigenetically modified sites [113]. BET inhibitors such as iBET 762 have recently shown promising efficacy in preclinical models for cancer treatments with clinical trials underway for breast and lung cancer treatments (NCT02964507, NCT01587703). Additional studies focused on iBET’s involvement in the tumor microenvironment. A 1-week short-term experiment showed an increase in helper T cells in the spleen of iBET-treated mice, but a 9-week study found a decrease in helper T cell population in the mammary glands of iBET-treated mice [79]. As a chemopreventive agent, oral treatment (60 mg/kg) of iBET significantly delayed mammary gland tumor formation by 3 weeks compared to control treatment in the MMTV-PyMT GEMM [79]. This 60 mg/kg dose was well tolerated with no signs of toxicity in mice [79].

*Vitamins and Micronutrients*. Vitamins and micronutrients have been found to impact various pathways in BC development. Among these, vitamin D3, folate, omega-3 polyunsaturated fatty acids, piperine, sulforaphane, indole-3-carbinol, epigallocatechine gallate, and quercetin along with curcumin have extensively been reviewed [114]. 

*Curcumin.* Curcumin is a hydroponic phenol derived from turmeric that has gained attention in the scientific community over the past decade for its effects as an anticancer agent. However, a large dose is required for effectiveness due to its low absorption. In vitro studies on curcumin have shown inhibition of migration, invasion, and angiogenesis, as well as induction of apoptosis and tumor suppressor genes [115]. Xenograft mouse models given oral doses of dendrosomal curcumin support these findings by showing reductions in tumor size, weight, and incidence compared to controls. Suppression of NF-kB, COX-2, and VEGF in these treated mice further supports curcumin as an anti-metastatic cancer agent [84]. Several clinical trials of curcumin in treatment of BC alone, or in combination with chemotherapy are currently active (NCT03980509, NCT03072992, NCT01740323, NCT01975363).

*RankL.* RankL is a protein secreted from osteoblasts which are most commonly known for their role in the formation of osteoclasts and bone remodeling. However, RankL is also expressed in the mammary gland during development and tumor formation [116,117,118]. Expression of RankL has been documented in BC cell lines and is associated with increased proliferation and poor survival in primary human BC [81]. Therefore, suppression of RankL is a promising new strategy in BC prevention. Xenograft mouse models given a RankL inhibitor, OPG-Fc, had increased tumor latency and reduced hyperplasia compared to the control mice. This was also supported in GEMMs treated with RankL-specific monoclonal antibodies [82].

### 4.3. Prophylactic Vaccines

In the late 2000s, there was a great interest in developing universal preventive vaccines against BC, in part spurred by the success of preventive vaccines against HPV-associated cervical cancer [119,120]. The Artemis project is the most well-known effort, supported by the National Breast Cancer Coalition and their initiative, to know how to eradicate BC by 2020 [121]. The scientific premise is to mount a specific immune response against mammary epithelial cells that would eliminate any neoplastic BC cells. As this initiative moved forward, the Artemis project Steering Committee in consultation with the FDA and other agencies raised concerns about clinical translation [122]. The main concern was the lack of a known antigen that would be expressed in 100% of the cells of all BC tumors. α-lactalbumin, mammaglobin-A, HER2, and survivin are among the best candidates, but studies suggest that none of them are represented in more than 80% of tumors [123]. More recently, the focus of the Artemis project has shifted from universal prophylaxis as primary prevention to targeted vaccination of specific subgroups of high-risk individuals for primary prevention or in specific patient subgroups to prevent metastatic recurrence [123]. Despite these caveats and challenges for a universal vaccine, progress has been made both in preclinical models and clinical trials that suggest the refinement and selection of the target population and combinatorial approach of vaccination, immunotherapy, and/or immune modulation could increase therapeutic efficacy in a preventive setting [119,120,124]. 

Preclinical models have shown a significant delay in tumor formation in mice vaccinated against α-lactalbumin [75,77] and HER2 [76,77]. α-lactalbumin is expressed at high levels exclusively during lactation in normal mammary epithelial cells, but α-lactalbumin expression is upregulated in about 50% of hormone receptor-positive and more than 70% of triple-negative BC tumors [119]. Subcutaneous injection of α-lactalbumin protein significantly delayed tumor formation in the MMTV-Erbb2 GEMM and elicited anti-α-lactalbumin-specific T cell response. However, α-lactalbumin vaccination can inhibit tumor growth but cannot delay tumor formation in more aggressive BC models [75,125]. In late 2020, a phase I clinical trial (NCT04674306) was started to investigate the safety of α-lactalbumin vaccine as adjuvant therapy in triple-negative BC patients at high risk of recurrence. HER2 is overexpressed in 20–30% of invasive BC tumors but expressed at very low levels in normal mammary epithelia. Intramuscular injections of virus-like particles studded with full-length HER2 or oncogenic variant Delta16HER2 extracellular domains in GEMMs significantly delayed or prevented tumor formation along with eliciting a robust anti-HER-2 production immune response [76]. A dendritic cell-mediated vaccine against HER2 also provided a tumor prevention effect in the full-length HER2-driven GEMM [77]. Mammaglobin A is a small glycoprotein that belongs to a family of epithelial secretory proteins. Mammaglobin A expression is upregulated in 40–80% of BCs [126]; unlike α-lactalbumin and HER2, mammaglobin A is more broadly expressed in normal mammary epithelial cells. Mammaglobin-A is exclusively expressed in BC cells, making it a suitable targeted protein for BC immunotherapy. Intramuscular injections of Mammaglobin A cDNA inhibit the growth of established tumors derived from multiple human BC cell lines in SCID-beige host animals and elicit a strong anti-Mammaglobin A T cell response [127,128]. Using this DNA vaccination approach (Figure 3A), anti-mammaglobin-specific T cells are readily detected in treated preclinical models, BC patients in phase 1 clinical trials, and metastatic BC patients in phase 1 clinical trials (NCT00807781) [89,129]. The results of this phase 1 clinical trial demonstrated the safety of this DNA vaccine and suggested a therapeutic effect based on extended progression-free survival of vaccinated participants [129]. These encouraging results have led to the ongoing phase 1b clinical trial (NCT02204098) investigating mammaglobin-A DNA in non-metastatic BC patients undergoing neoadjuvant endocrine therapy or chemotherapy [126]. However, this DNA vaccination approach has not been tested in a primary prevention setting in either preclinical models or in clinical trials. 

## 5. Local Approaches for Primary Prevention in Preclinical Model and Clinical Trials

Each of the two mammary glands of a woman contains 8–12 ductal trees with the main duct of each tree opening at the nipple orifice [27,28]. The main function of these ductal tree systems is to produce, express, and deliver milk during lactation. Alveolar cells arranged in lobules at the ends of the ductal tree produce and secrete milk components into ductules that join into larger ducts and eventually the main duct to transport the milk outwardly. Breast carcinoma predominantly arises from the terminal ductal lobular units (TDLUs) [130,131] within a single ductal tree [26]. Remodeling of TDLUs during an individual’s childbearing years and aging influences cancer risk. Epidemiological and genetic susceptibility studies suggest that the number and differentiation state of mammary epithelial cells correlated with BC risk; for example, increased lobular involution (fewer TDLU) as a result of the aging process is protective against BC [132,133,134]. The opening at the nipple into the ductal tree offers a unique opportunity for BC prevention by directly targeting the epithelial cells of the ductal tree. In 2002, Sivaraman et al. introduced the concept of local ablation for primary prevention and local treatment of early BC by intraductal (ID) delivery of a suicidal gene system [97]. Although this study was unsuccessful in reducing tumor incidence, it unveiled a new approach and local strategy for BC prevention. Within the last twenty years, several groups have built on this local treatment concept that focuses on the ductal tree as a functional unit consisting of individual pre-malignant and/or malignant cells for primary prevention of BC (Table 3). A common theme of many of these local approaches is the delivery of the active compound at a lower dose that achieves the same efficacy on targeted cells and/or minimizes the undesired side effects of systemic exposure (Figure 1D,E, Figure 2B–D and Figure 3B–E). 

### 5.1. Gene Therapy

Gene therapy utilizes DNA or RNA sequences to modify gene activity or expression. Some gene therapy approaches such as viral delivery of DNA or genetic editing of a patient’s genome may offer a more permanent treatment compared to continuous and multi-dose treatment of chemoprevention and endocrine therapies. Here, we discuss the ID delivery of components of different gene therapy platforms.

*Suicide gene therapy.* An adenoviral vector was used to intraductally deliver a thymidine kinase-based suicidal gene system in an N-methyl-N-nitrosourea (MNU)-induced rat model of BC [97]. The therapeutic intent was to prevent tumor formation by ablating the proliferating cells of the terminal end buds akin to human TDLUs. However, treated rats paradoxically developed tumors earlier than control rats despite the high transduction efficiency, high thymidine kinase expression, and 50–90% epithelial ablation by suicidal gene activation.

*RNAi-based gene silencing.* ID delivery of lipidoid nanoparticles containing small interfering RNAs (siRNAs) against Hox1A in the C3(1)-TAg GEMM resulted in a significant delay in tumor latency [98]. Hox1A is a homeodomain transcription factor involved in autocrine growth hormone dysregulation and was identified as an early driver of mammary cell carcinogenesis [98]. Knocking down *Hox1A* mRNA maintained hormone receptor status and reduced the proliferative rate of pre-malignant cells [85]. Remarkably, researchers were able to sustain this knockdown effect by repeatedly cannulating and delivering 20 μL of therapeutic siRNAs in nine biweekly procedures without causing any damage or perforation to ductal trees. This study also demonstrated the efficiency of in vivo transfection of the ductal tree using components of RNAi technology and opened the possibility of introducing components of other gene-silencing or -editing technologies.

*CRISPR-based genetic editing.* CRISPR/Cas9 is a versatile gene-editing tool to produce loss of function or knockout alleles in in vitro cell line systems and in vivo animal models [135]. CRISPR technology has already reached the clinical trial stage. Although the majority of these trials involved ex vivo editing of hematopoietic stem cells and T cells, in vivo editing of liver cells with a systemic delivery approach and eye cells with a local delivery approach have been successful [135]. ID delivery of the CRISPR system to target mammary epithelial cells has been applied to develop new models of BC and to study cooperative interactions of multiple driver genes [136,137,138,139,140]. ID delivery of CRISPR components (Cas9 mRNA and guide RNA) via lentiviral vector achieved up to 21.5% editing efficacy of the target gene [137]. An alternative approach to minimize the immunogenicity of de novo Cas9 expression is to intraductally deliver the guide RNA in a lentiviral vector in animals endogenously expressing Cas9 [137,140]. Up to 32.6% editing efficacy of the target gene may be achievable using this modified system [140]. Although further development of this technology will be required to identify a safe delivery system with an improved editing efficacy, ID delivery of CRISPR or similar system for editing and/or inactivation of early driver genes of mammary carcinogenesis such as estrogen receptor α (*ESR1*), *HOXA1*, and/or *EGFR* may offer new opportunities for BC prevention. It is tempting to speculate that such genetic editing approaches to dampen mitogenic activity of ER signaling may provide a local alternative to systemic hormonal control but with minimal adverse effects (Figure 2D). 

### 5.2. Local Hormone Therapy

Different approaches are being investigated to minimize side effects of systemic hormone therapy. More readily available than the above-mentioned genetic editing approach is the local delivery of a selective estrogen receptor modulator (SERM) or degrader (SERD). Local delivery methods include topical transdermal gel application, ID injection, and slow-releasing drug implants (Figure 2B,C). 

*Transdermal Gel for hormone therapy.* Topical application to the skin of the breast area of a gel containing active metabolite 4-hydroxy tamoxifen achieved a high local concentration of this SERM [141,142]. A phase 2 clinical trial (NCT00952731) divided 27 women diagnosed with DCIS into two groups to receive transdermal gel application or tamoxifen orally for six to ten weeks before surgery. Due to the early stage of the disease, the proliferation index Ki-67 was used as the endpoint rather than tumor incidence. Compared to pre-treatment tissue samples, the proliferative index was reduced by 52% in the transdermal gel group and by 61% in the tamoxifen group. At a local level, both groups had the same concentration of this SERM. Systemically, the transdermal gel group had reduced plasma concentrations of the SERM compared to the tamoxifen group. However, side effects such as hot flashes did not significantly differ between the two groups [59]. The timeframe of this study limited the assessment of long-term side effects, but this is currently being evaluated in a phase 2 clinical trial (NCT03063619) whose primary objective is to determine risk reduction in BC by transdermal gel application (Table 1, Figure 2B). An alternative study was conducted using z-form endoxifen, a tamoxifen metabolite with the highest affinity to the ER, to study BC prevention [71]. Groups of 30 women self-applied 10 mg, 20 mg, or a placebo topically to each breast daily for three months and were monitored for breast density changes, systemic side effects, and plasma concentration of endoxifen [71]. Before the end of the three-month period, there was a significant decrease in mammographic density of women applying 20 mg endoxifen which was not seen in the 10 mg treatment group. Dose-dependent plasma concentrations of endoxifen were observed in the treatment groups without systemic side effects [71]. However, severe skin reactions occurred in both treatment groups which caused almost all women (58 out of 60) to prematurely discontinue the gel application and no therapeutic window was identified [58]. Therefore, endoxifen has the potential to reduce BC incidence but requires further studies due to severe skin toxicity.

*Intraductal hormone therapy.* ID delivery of tamoxifen did not provide a protective effect on an MNU-induced rat model of BC most likely due to the lack of active metabolite production in the liver. However, ID delivery of 4-hydroxytamoxifen provided a protective effect comparable to subcutaneous injection of tamoxifen [91]. Fulvestrant acts as a selective estrogen receptor degrader (SERD). Fulvestrant binds with a higher affinity to the ER than tamoxifen and unlike tamoxifen is a pure antagonist by degrading ER upon binding [143]. ID delivery of fulvestrant in a mammary intraductal (MIND) ER+ MCF7 xenograft model provided superior protection than intramuscular injection, whereas ID delivery of fulvestrant provided a protective effect comparable to intramuscular injection in an MNU-induced rat model [95]. Locally delivered fulvestrant was more effective at inhibiting cell proliferation, angiogenesis, and decreasing ER expression. In both mouse and rat models, the total amount of fulvestrant received per animal was the same by ID delivery (fragmented dosing per ductal tree) and by intramuscular injection [95]. A phase 2 clinical trial (NCT02540330) was initiated to investigate the pharmacokinetics and local and systemic side effects of ID delivery compared to intramuscular injection (Table 1, Figure 2C). However, due to a business decision, the study was terminated after the recruitment of only three participants. 

*Slow-release implant of hormone therapy.* To expand on the application of fulvestrant as a local long-term prevention treatment, bilateral fulvestrant-loaded silastic tubing was subcutaneously implanted next to the abdominal mammary glands of NSG mice [96]. One week after implantation, the mice were orthotopically injected with MCF7 cells in both fat pads of the abdominal glands [96]. A significant tumor growth reduction was observed in the fulvestrant-loaded silastic tubing group compared to vehicle control treatment [96]. Importantly, the treatment efficacy of fulvestrant-loaded implants was very similar to weekly treatment with subcutaneous injection of fulvestrant. This is an encouraging study to consider for local drug delivery to the mammary gland. However, more investigations are needed to have a more controlled and homogenous release of fulvestrant or other drugs throughout the entire mammary gland since a much more pronounced decrease in the Ki-67 proliferation marker was measured in tumor areas adjacent to the tubing implant [96]. 

### 5.3. Intraductal Chemotherapy and Targeted Treatments

Several research groups have used ID delivery of cytotoxic compounds, targeted agents, and/or targeted particles for both primary prevention of BC (Table 3, Figure 3C–E) and preclinical models for local disease control [144].

*Infusion of cytotoxic agents.* Cytotoxic compounds such as pegylated liposomal doxorubicin, carboplatin, or paclitaxel when ID delivered in the chemically induced (MNU) rat model significantly reduced tumor incidence [90,91,92]. Similarly, liposomal doxorubicin showed greater therapeutic efficacy at reducing tumor incidence when ID was delivered rather than systemic administration in the genetic MMTV-Erbb2 mouse model [91]. Several groups have already tested the feasibility of this approach in first-human clinical studies. Stearns et al. 2011 reported an 88% success rate by administering pegylated liposomal doxorubicin into one ductal tree per patient [92]. Love et al. 2013 reported a 96.6% success rate in administering pegylated liposomal doxorubicin into 5–8 ductal trees per patient) [145]. These studies provide strong support for the translational feasibility of ID delivery of cytotoxic or other solutions. Unfortunately, local cytotoxic treatment with pegylated liposomal doxorubicin, 5-fluorouracil, and/or cisplatin can induce tumors with long latency in non-transgenic animals [89,94,146]. This result has diminished enthusiasm for such local chemotherapy delivery unless it is needed to minimize systemic exposure [89,146].

*Infusion of targeted agents.* Pseudomonas exotoxin has been engineered to serve as an anti-cancer agent thanks to its effect on protein translation and cell death [147]. However, systemic delivery of this exotoxin has undesired effects that lead to inflammation and vascular leakage. Transferrin receptor is overexpressed in the majority of BC cells, in many cases starting at a preneoplastic stage. To harness the anti-tumoral effect of this exotoxin, but to keep it contained locally within the ductal tree, Wang et al. 2022 fused a monoclonal antibody targeting the human transferrin receptor to a 40 kDa fragment of this exotoxin [100]. ID delivery of this antibody–toxin conjugate shows therapeutic efficacy in HER2+ MIND models of MCF7 and SUM225 human BC cells that recreate the early stage of DCIS lesions [100]. A similar approach was utilized to deliver an a-emitter radionuclide, as the killing agent, conjugated to a HER2-targeting antibody (trastuzumab). a-emitters can cause irreparable double-strand DNA damage that leads to cell death [148]. The radionuclide used in this study was ^225^Ac with a half-life of 9.9 days [148]. ID delivery of this antibody–^225^Ac conjugate shows therapeutic efficacy in the HER2+ MIND model of SUM225 BC cells [99]. However, this treatment can induce tumor formation in the mammary gland or lung due to sustained radiation exposure [99]. As the field of radionuclides and conjugation chemistry continues to mature [148,149], the use of a-emitters with shorter half-life (<12 h) such as ^211^At and ^212^Pb could minimize the undesirable iatrogenic effects of trastuzumab-^225^Ac. Although these studies were proof-of-concept for clinical treatment of DCIS due to the requirement of transferrin receptor or HER2 expression, conjugating these cell-killing agents to other antibodies, peptides, or targeting moieties could expand their applications to primary prevention of BC and/or local control of uninvolved ductal trees in DCIS-affected breast. Similarly, antibody–drug conjugates [150] such as trastuzumab deruxtecan for HER2-low BC and sacituzumab govitecan for triple-negative BC could be considered for primary prevention.

*Ductal tree ablation*: We have been intrigued by the concept of a universal prophylactic intervention for several years. Our approach aims at combining the effectiveness of prophylactic mastectomy, with the universality of an ideal vaccine and delivery method of ID chemotherapy (Figure 1D and Figure 3B). We are investigating different chemical and thermal ablation approaches to locally kill mammary epithelial cells while causing minimal collateral tissue damage and side effects. We have extensively studied ethanol as a cell-killing solution for chemically ablating the ductal tree [101,151,152,153,154,155]. In clinical settings for ablation or sclerotherapy, tens of milliliters of 95–100% ethanol (EtOH) can be locally delivered to the target area [156,157,158,159,160,161,162,163,164,165,166,167,168,169,170]. In some procedures up to 50 mL of EtOH can be administered per session, indicating its low toxicity [156,166]. We have demonstrated in preclinical rodent models the feasibility of ablating the entire ductal tree system before epithelial cells become malignant [101,152,153]. Our study in the C3(1)-TAg GEMM showed that ID injection of 70% EtOH significantly delayed tumor formation and reduced tumor incidence [101]. ID injection of 70% EtOH provides similar or superior tumor risk reduction compared to other prevention interventions in this or similar GEMMs (Table 3; refs [91,98]). This chemical ablation approach has advantages over other ID approaches for clinical translation. It would be a one-time treatment in contrast to other ID approaches that require repeated administration of active agents which can be a challenge, especially for chemotherapy agents that may compromise ductal tree structure and leak out of intended targeted area in later cycles. Currently, there are no reports linking clinical uses of EtOH to iatrogenic cancer, but this is a concern for other local treatments that cause DNA damage such as radioimmunotherapy and chemotherapy (Figure 3D,E; refs [94,99,146,171]). The International Agency for Research on Cancer considers EtOH to be carcinogenic to humans [172]; however, this is based on chronic exposure to EtOH as an alcoholic beverage. EtOH is metabolized into acetaldehyde, a toxic chemical that causes DNA damage and DNA-protein cross-linking. This is considered a main contributing mechanism for EtOH-induced cancers but the exact molecular mechanism(s) of EtOH and increased cancer risk is not fully established [172]. No significant DNA damage was seen with acute EtOH exposure in mice [173,174]. Our studies showed no evidence of iatrogenic effects of EtOH injections in non-transgenic mice after 22 months of follow-up [101]. There are also some unique challenges with this chemical ablation approach. Image guidance will be required for precisely infusing each ductal tree with the appropriate volume. We and others have used different contrast agents for in vivo imaging of the ductal tree in rodents [101,151,152,153,154,175,176] and rabbits [177] after ID infusion. Controlling the diffusion of EtOH outside the ductal tree will be required to further minimize collateral tissue damage [101]. The use of ethyl cellulose as a gelling agent to limit ethanol diffusion has been reported for clinical treatment of venous malformation and in preclinical models of BC, cervical cancer, and liver cancer [152,159,160,161,162,163,164,165,178]. We have shown that ethyl cellulose is compatible with a 70% EtOH ablative solution and with imaging contrast agents such as tantalum oxide nanoparticles in both mouse and rat models [152,153].

Similar to chemical ablation, thermal ablation aims to target the ductal tree with minimal collateral damage and improved cosmesis. This technique uses physical energy to raise or lower the temperature to internally target local tumors instead of surgical removal [179]. Procedures such as microbubble solutions with high-intensity ultrasound or an iron rod nanoparticle solution coupled with a magnetic field have been previously used in cancer models such as pancreatic xenograft mouse models [180,181]. These thermal ablation techniques, coupled with their imaging abilities, may be applied toward targeting epithelial cells within the ductal tree for BC prevention. 

## 6. Conclusions and Future Directions

BC is the leading cause of new cancer cases and is the second leading cause of cancer-related deaths in women. The limited, approved prevention interventions available for high-risk individuals can have severe and long-term side effects that deter many women from making proactive choices. Regrettably, BC prevention is an area of translational research that is currently underfunded by federal agencies [182]. Therefore, emerging and novel approaches for the primary prevention of BC that provide the same or superior protection while minimizing the side effects of current interventions should be pursued and prioritized. These approaches seek to eradicate BC and align well with NIH’s All of US Research Program—a nationwide initiative for precision health interventions that proactively prevent rather than treat disease in high-risk individuals. We provide a comprehensive review of emerging approaches for BC prevention based on non-modifiable risk factors that put women at a higher risk of developing BC. Virtually all of these studies were performed in rodent models of BC, which have some limitations and challenges for direct translation to at-risk individuals. Additional scalability and validation studies in larger animal models will be an important step in bridging the gap between discovery and clinical evaluation. Although currently underutilized, rabbit studies should be considered as an appropriate intermediate model. Evolutionarily, anatomically, and physiologically, rabbit mammary glands are more similar to humans than those of rodent models or other large animals such as cows and sheep [183,184]. Female rabbits have four pairs of mammary glands each containing four ductal trees [177], which can be cannulated for ID injection [177,185,186,187,188,189] using a procedure very similar to ID administration of contrast agents in clinical ductography [190,191] and chemotherapeutic agents in first-in-human clinical research [92,145]. Alternatively, or complementary to validation studies in larger animals, judicious determination based on the scientific rigor and clinical feasibility of these emerging approaches should be applied to prioritize those interventions more likely to have an impact on primary prevention of BC. 

## Figures and Tables

**Figure 1 cancers-16-00248-f001:**
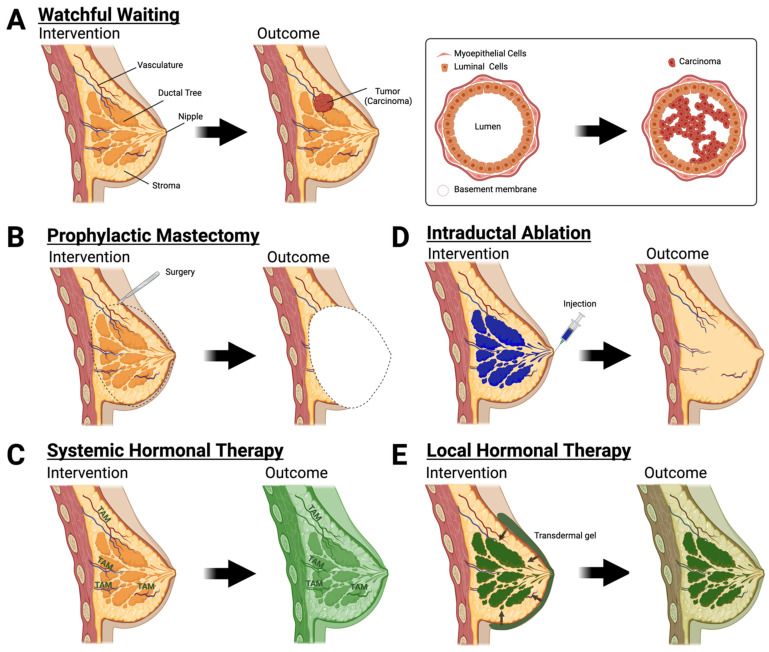
Evidence-based and investigational approaches for primary prevention. (**A**) Watchful waiting with no preventative outcome. (**B**) Bilateral prophylactic mastectomy is an effective risk-reducing surgery. (**C**) Hormonal therapy such as tamoxifen (TAM) or raloxifene can reduce risk but with high systemic exposure (dark green shading). (**D**,**E**) Intraductal approaches for breast cancer prevention. (**D**) Intraductal injections ablate epithelial cells leaving the breast stroma intact. (**E**) Local hormonal therapy with moderate systemic exposure (light green shading).

**Figure 2 cancers-16-00248-f002:**
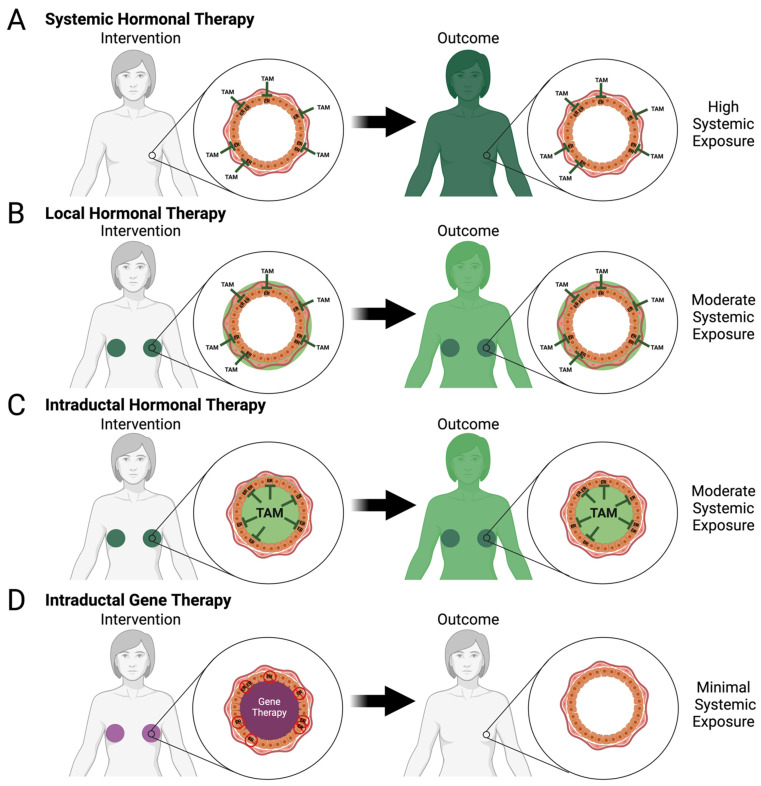
Systemic and local hormone therapy approaches for breast cancer prevention. (**A**) Systemic hormonal therapy with high systemic exposure (dark green shading). (**B**) Local hormonal therapy with concentrated exposure to the ductal tree (dark green shading) and moderate systemic exposure to the body (light green shading). (**C**,**D**) Intraductal delivery directly to the ductal tree by hormonal therapy with moderate systemic exposure (light green shading) or gene therapy with minimal systemic exposure (no shading).

**Figure 3 cancers-16-00248-f003:**
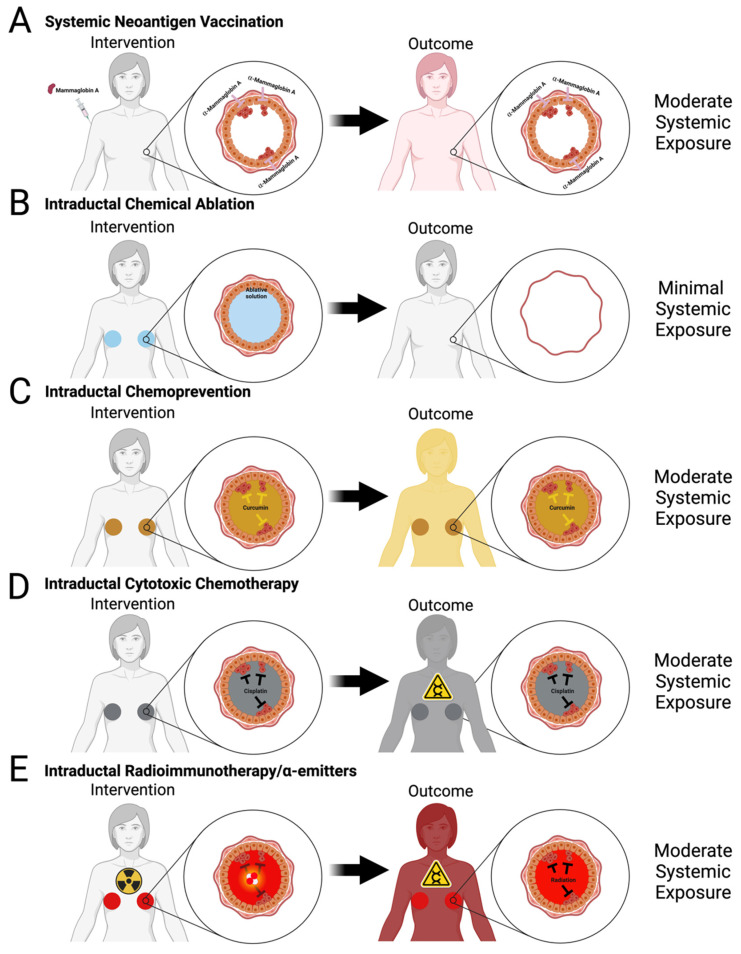
Systemic vaccination and intraductal approaches for breast cancer prevention. (**A**) Vaccines target cancer cell-expressed proteins and mount an immune response against (pre)malignant epithelial cells. (**B**–**E**) Intraductal injections provide direct delivery to epithelial cells that may become malignant. This form of delivery can be used with multiple solutions including ablative solutions (ethanol in (**B**)), chemopreventives (curcumin in (**C**)), cytotoxic chemotherapeutics (cisplatin in (**D**)), or radioimmunotherapy/α-emitters in (**E**). Intraductal chemical ablation and vaccines only require 1–2 injections and have minimal to moderate systemic exposure, unlike cytotoxic chemotherapeutics/radioimmunotherapeutics with potential iatrogenic carcinogenesis.

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
