# Peer review of "Systemic and Local Strategies for Primary Prevention of Breast Cancer"

_cancers, 2024, doi:10.3390/cancers16020248_

Round 1

Reviewer 1 Report

Comments and Suggestions for Authors

Interesting and useful topic in the field of breast cancer. Well organized paper targeting the main points of the aim. It should be very useful to add a brief critical overview of antiangiogenic/antivascular therapies as preventive therapies in benign breast lesions. Also a brief chapter with growth factors evolution during benign breast lesions with potential to become malignant should be inserted. All related to various genetic predisposition variabilities. 

Comments on the Quality of English Language

Minor revisions.

Author Response

Reviewer comments: Breast Cancer Review

Dear Editorial Team,

We appreciate the overall positive comments and constructive critiques provided by the reviewers. We address reviewers’ concerns point by point below and we have accordingly made revisions in throughout the main text and figures.

These revisions are indicated with markup tracked changes. We hope that these revisions and additional information provided are considered responsive to reviewers’ report as well as editorial team’s modifications, and satisfy both the reviewers and the editorial team.

Reviewer 1:

Comments and Suggestions for Authors

Interesting and useful topic in the field of breast cancer. Well organized paper targeting the main points of the aim. It should be very useful to add a brief critical overview of antiangiogenic/antivascular therapies as preventive therapies in benign breast lesions. Also a brief chapter with growth factors evolution during benign breast lesions with potential to become malignant should be inserted. All related to various genetic predisposition variabilities. 

We agree that these are interesting topics, but we believe that they relate more to cancer interception and local treatment. The focus on this review is primary prevention and not interception. We are not aware of recent reviews covering these topics and to introduce and describe appropriately in this manuscript would distract and deviate from the intended scope.

Reviewer 2 Report

Comments and Suggestions for Authors

The submitted review entitled: Systemic and Local Strategies for Primary Prevention of Breast Cancer, presents a very interesting, well written, logically sequenced and highly informative study on the preventive methods for breast cancer. Yet, the following points are suggested to be addressed before further steps:

1-While the figures are very useful and summative, but they need further explanation and numbering of each part.

2-Figure 3 needs to be in higher resolution

3-It is suggested to include the breast cancer preventive role of other Vitamins (i.e. B, E, etc) and Micromutrrients (Zinc, etc).

4-In this context, it is suggested to briefly discuss the possible preventive role of antibody-drug conjugates.

Author Response

Reviewer comments: Breast Cancer Review

Dear Editorial Team,

We appreciate the overall positive comments and constructive critiques provided by the reviewers. We address reviewers’ concerns point by point below and we have accordingly made revisions in throughout the main text and figures.

These revisions are indicated with markup tracked changes. We hope that these revisions and additional information provided are considered responsive to reviewers’ report as well as editorial team’s modifications, and satisfy both the reviewers and the editorial team.

Reviewer 2:

Comments and Suggestions for Authors

The submitted review entitled: Systemic and Local Strategies for Primary Prevention of Breast Cancer, presents a very interesting, well written, logically sequenced and highly informative study on the preventive methods for breast cancer. Yet, the following points are suggested to be addressed before further steps:

1-While the figures are very useful and summative, but they need further explanation and numbering of each part.

We have made this correction.

2-Figure 3 needs to be in higher resolution

We have made this correction.

3-It is suggested to include the breast cancer preventive role of other Vitamins (i.e. B, E, etc) and Micromutrrients (Zinc, etc).

Vitamins and Micronutrients have recently been extensively reviewed for breast cancer prevention by Mokbel & Mokbel in 2019. We have included a brief paragraph directing readers to this review. Vitamins and micronutrients can be considered part of diet or dietary intervention which are more related to modifiable risks since the review focuses more on non-modifiable risks, we hesitate to elaborate much more here.

4-In this context, it is suggested to briefly discuss the possible preventive role of antibody-drug conjugates.

We are not aware of the use of ADC in a prevention setting. We had a section on targeted agents (trastuzumab-225Ac and transferrin receptor-exotoxin) and implicitly alluded to adding other payloads. We now briefly expand this section to explicitly include ADC and a recent review on their treatment application in breast cancer.

Reviewer 3 Report

Comments and Suggestions for Authors

The authors present a review of risk reduction options for breast cancer. 

Specific comments

1.       Abstract ‘. Despite, evidence-based effectiveness of these interventions, fewer than 30% of high-risk individuals opt for bilateral prophylactic mastectomy and fewer than 10% opt for anti-estrogen therapy.’ -These figures do not reflect the complete literature. Long term uptake of BRRM can be as high as 50% for BRCA1 and newly identified high risk women can have uptakes for anti-estrogen therapy of >60%. Please rewrite

2.       ‘. Likely, these strategies would be initially tested in high-risk individuals, but may be applicable to moderate- and low-risk women’ -do you really mean ‘low risk’ don’t you mean ‘average’ risk?

3.       ‘. With the number of new diagnoses still on the 35 rise, 1 in 8 women have now a lifelong risk of developing BC.’ -This needs rephrasing. Al women have a risk not just 1 in 8

4.       ‘. Therefore, there is a need to develop new strategies for primary prevention with a focus on high-risk individuals, but with strategies that can also be applied to moderate- and low-risk individuals.’ -As above why ‘low risk’?

5.       ‘. Cumulative exposure of breast tissue to estrogen is an important risk factor(s).. ‘ -delete ‘s’

6.       ‘, and other non-modifiable components such as age at menarche and menopause [2].’ -menopause can be modifiable as many BRCA1/2 carriers have oophorectomy

7.       ‘. There are also non-modifiable risk factors related to personal history of radiation therapy to the chest, increased breast density’ -Breast density can be reduced by taking tamoxifen

8.       ‘Women with other genetic predispositions including (mutation) in CDH1…’ -Most journals now use pathogenic variants

9.       ‘. Women with other genetic predisposition including mutations in ATM and CHK2 or who carry risk-associated alleles for one or more of 92 susceptibility genes are generally considered moderate-risk individuals [10].’ There are now more than 300 SNPs but carrying one risk allele alone will NOT make someone moderate risk. Only a proportion with an odds ratio above 1.5 will be moderate risk depending on other risk factors

10.   ‘. Genetic counseling based on 20-gene panels captures most frequent mutations’ -Why 20 there are only 13 genes clearly linked to actionable risk see CARRIERS and BRIDGES studies NEJM 2021

11.   ‘Due to adverse side effects, recommendation for these interventions is generally restricted to high-risk individuals.’ -You need to define in the intro what you mean by high risk. In Europe this is a LT risk of 30% or higher

12.   ‘Prophylactic mastectomy. Prophylactic mastectomy is a highly invasive procedure can be executed as a total mastectomy, can …’ -Most people and journals now refer to this as risk reducing mastectomy

13.   ‘…this invasive surgery removes the epithelial cells, the intended target, from which breast carcinomas arise’ -You have already said invasive once

14.   ‘. Regrettably, > 25% of patients suffer from post-mastectomy chronic pain syndrome [23, 24] and/or other side effects that affect quality of life [25, 26].’ -You should only use studies that assessed this in true prevention patients. In cancer patents other factors could influence this

15.   There is no mention of level of risk reduction from bilateral mastectomy

16.   ‘Salpingo-oophorectomy. Surgical removal of the fallopian tubes and ovaries is a commonly recommended BC prevention of premenopausal BRCA1/2 mutation carries with a risk reduction of up to 50% [27].’ –This paper is an outlier and the 50% figure is from older publications not this one. Most recent publications with left censoring do not suggest a benefit in BRCA1 and a later benefit in BRCA2

17.   ‘Immediately following salpingo-oophorectomy, women will experience menopausal symptoms from the removal of estrogen and progesterone hormones.’ This is not inevitable. Many women have no symptoms even when carried out pre-menopausal. Post menopause oophorectomy will not usually give any symptoms but will not reduce breast cancer risk

18.   ‘Selective estrogen receptor modulators (SERMs), tamoxifen and raloxifene, are commonly used in the treatment of BC, osteoporosis, and postmenopausal symptoms and are the only two FDA approved compounds for primary prevention of BC [4, 32].’ -What about aromatase inhibitors? Exemestane in the USA?? Anastrozole has just been approved by the MHRA in UK. Also SERMS are NOT used for post menopausal symptoms as they make them worse!

19.   ‘Additional studies of tamoxifen for *preventive agent reported these adverse effects as major factors contributing to discontinuation of treatment, especially among women taking tamoxifen for primary prevention as opposed to adjuvant therapy for BC [41, 42].’ -the section on tamoxifen is a joke. There is much more important long term data from the IBIS trials that is completely ignored. NASBP closed after only 4 years follow up and has no useful long term data

20.   ‘Direct comparison of tamoxifen and raloxifene revealed equal risk reduction of invasive BC, but endometrial cancer in postmenopausal women, thromboembolic events, and stroke occurred in both groups [45].’-What about longer term follow up? -this showed tamoxifen was better for risk reduction?

Comments on the Quality of English Language

Quite a lot of basic errors

Author Response

Reviewer comments: Breast Cancer Review

Dear Editorial Team,

We appreciate the overall positive comments and constructive critiques provided by the reviewers. We address reviewers’ concerns point by point below and we have accordingly made revisions in throughout the main text and figures.

These revisions are indicated with markup tracked changes. We hope that these revisions and additional information provided are considered responsive to reviewers’ report as well as editorial team’s modifications, and satisfy both the reviewers and the editorial team.

Reviewer 3:

Comments and Suggestions for Authors

The authors present a review of risk reduction options for breast cancer. 

Specific comments

  1. Abstract ‘. Despite, evidence-based effectiveness of these interventions, fewer than 30% of high-risk individuals opt for bilateral prophylactic mastectomy and fewer than 10% opt for anti-estrogen therapy.’ -These figures do not reflect the complete literature. Long term uptake of BRRM can be as high as 50% for BRCA1 and newly identified high risk women can have uptakes for anti-estrogen therapy of >60%. Please rewrite

We revised those numbers and added new references to support the BRRM numbers, however, we cannot find a recent supporting the >60% risk reduction of anti-estrogen therapy. If the reviewer would kindly provide that reference we will add to manuscript.

  1. ‘. Likely, these strategies would be initially tested in high-risk individuals, but may be applicable to moderate- and low-risk women’ -do you really mean ‘low risk’ don’t you mean ‘average’ risk?

We have made this correction. We revise the text and use “average” instead of “low” risk for clarity. We define average is at 12.5%, moderate 15 to 20% and high risk > 20% . We added a reference from Michaels et al 2023 that states high risk women is >20% aligning with ASCO and ACOG guidelines (Please see reference 2)

Reference added:

 Michaels E, Worthington RO, Rusiecki J: Breast Cancer: Risk Assessment, Screening, and Primary Prevention. Med Clin North Am 2023, 107(2):271-284.

  1. ‘. With the number of new diagnoses still on the rise, 1 in 8 women have now a lifelong risk of developing BC.’ -This needs rephrasing. All women have a risk not just 1 in 8

We have made this correction. This is a good catch; we have clarified language about actual numbers and risk of developing breast cancer.

  1. ‘. Therefore, there is a need to develop new strategies for primary prevention with a focus on high-risk individuals, but with strategies that can also be applied to moderate- and low-risk individuals.’ -As above why ‘low risk’?

We have made this correction. As we described above, we consistently use now “average” risk.

  1. ‘. Cumulative exposure of breast tissue to estrogen is an important risk factor(s).. ‘ -delete ‘s’

We have made this correction.

  1. ‘, and other non-modifiable components such as age at menarche and menopause [2].’ -menopause can be modifiable as many BRCA1/2 carriers have oophorectomy

We have made this correction. This is a good point. We qualified this is “natural” menarche and menopause.

  1. ‘. There are also non-modifiable risk factors related to personal history of radiation therapy to the chest, increased breast density’ -Breast density can be reduced by taking tamoxifen

Yes, we agree with the review that both non-modifiable and modifiable conditions can contribute to breast density including hormonal treatment and low BMI. We have removed “increased breast density” as a solely non-modifiable risk to avoid confusion.

  1. ‘Women with other genetic predispositions including (mutation) in CDH1…’ -Most journals now use pathogenic variants

Pathogenic variants imply, at least to us, causality. There are some variants of unknown significance. Without functional demonstration, we find mutation a more neutral term since not all DNA variants will have an impact on risk (e.g, mutation in intron, enhancer, promoter). However, we are open to changing this if reviewer finds this adds value to the review.

  1. ‘. Women with other genetic predisposition including mutations in ATM and CHK2 or who carry risk-associated alleles for one or more of 92 susceptibility genes are generally considered moderate-risk individuals [10].’ There are now more than 300 SNPs but carrying one risk allele alone will NOT make someone moderate risk. Only a proportion with an odds ratio above 1.5 will be moderate risk depending on other risk factors

Yes, we agree that this is a compounded risk of the 92 gene panel and more than one allele associated with higher risk is needed to consider the patient at a moderate rather than average risk. We have accordingly modified the text.

  1. ‘. Genetic counseling based on 20-gene panels captures most frequent mutations’ -Why 20 there are only 13 genes clearly linked to actionable risk see CARRIERS and BRIDGES studies NEJM 2021

We use the 20 plus gene panel based on references:

  1. Rosenthal ET, Evans B, Kidd J, Brown K, Gorringe H, van Orman M, Manley S: Increased Identification of Candidates for High-Risk Breast Cancer Screening Through Expanded Genetic Testing. J Am Coll Radiol 2017, 14(4):561-568.
  2. Gallagher S, Hughes E, Wagner S, Tshiaba P, Rosenthal E, Roa BB, Kurian AW, Domchek SM, Garber J, Lancaster J et al: Association of a Polygenic Risk Score With Breast Cancer Among Women Carriers of High- and Moderate-Risk Breast Cancer Genes. JAMA Netw Open 2020, 3(7):e208501.

The review is correct that the CARRIERS and BRIDGES studies do not find association with risk for all 30 plus predisposition genes. The predisposition genes in the  CARRIERS and BRIDGES studies overlap with the ones that we provide on ref 5 and 6. Both studies provide explanations as to why not all predisposition genes were linked to an elevated risk in the studied populations.

The BRIDGES study find evidence of elevated risk with 15 genes (https://www.nejm.org/doi/full/10.1056/nejmoa1913948) and CARRIERS study with 12 genes (https://www.nejm.org/doi/full/10.1056/NEJMoa2005936). We interpret these results like the review as the most informative genes to assess risk, but unlike the reviewer we don’t consider that this means that those other genes are not linked to higher risk in very specific enriched subpopulations that were not adequately powered to be detected in these studies.

We have now added the CARRIERS and BRIDGES studies after ref 5 and 6. As these studies as pointed out by reviewer are more recent.

  1. ‘Due to adverse side effects, recommendation for these interventions is generally restricted to high-risk individuals.’ -You need to define in the intro what you mean by high risk. In Europe this is a LT risk of 30% or higher

We now state more clearly both in abstract and introduction the risk associated with average, moderate and high individual.

  1. ‘Prophylactic mastectomy. Prophylactic mastectomy is a highly invasive procedure can be executed as a total mastectomy, can …’ -Most people and journals now refer to this as risk reducing mastectomy

We agree that prophylactic mastectomy and risk-reducing mastectomy have same meaning.  In the context we use it, (dual) prophylactic mastectomy is more appropriate since performing a risk-reducing mastectomy in high-risk patients seems a bit redundant. We do, however, introduce thanks to reviewer’s comments the language of risk-reducing procedure or intervention as appropriate throughout the text.

  1. ‘…this invasive surgery removes the epithelial cells, the intended target, from which breast carcinomas arise’ -You have already said invasive once

We have made this correction.

  1. ‘. Regrettably, > 25% of patients suffer from post-mastectomy chronic pain syndrome [23, 24] and/or other side effects that affect quality of life [25, 26].’ -You should only use studies that assessed this in true prevention patients. In cancer patents other factors could influence this

We  have made this correction.  We agree that is more appropriate to use a refence in the context of prevention to avoid other confounders.

Gahm J, Wickman M, Brandberg Y: Bilateral prophylactic mastectomy in women with inherited risk of breast cancer--prevalence of pain and discomfort, impact on sexuality, quality of life and feelings of regret two years after surgery. Breast 2010, 19(6):462-469.

  1. There is no mention of level of risk reduction from bilateral mastectomy

In the context of this review prophylactic mastectomy include removal of both breast since a tumor could eventually for in either of the breast, risk reduction benefit is maximal with both breast removed. We explicitly mention that this a dual prophylactic mastectomy.

  1. ‘Salpingo-oophorectomy. Surgical removal of the fallopian tubes and ovaries is a commonly recommended BC prevention of premenopausal BRCA1/2 mutation carries with a risk reduction of up to 50% [27].’ –This paper is an outlier and the 50% figure is from older publications not this one. Most recent publications with left censoring do not suggest a benefit in BRCA1 and a later benefit in BRCA2

We have made this correction. We updated references and discuss on text controversy surrounding previous study. References 28-31 are new that are updated data that account of time varying covariate.

  1. ‘Immediately following salpingo-oophorectomy, women will experience menopausal symptoms from the removal of estrogen and progesterone hormones.’ This is not inevitable. Many women have no symptoms even when carried out pre-menopausal. Post menopause oophorectomy will not usually give any symptoms but will not reduce breast cancer risk

We have made this correction. Please see reference 31.

  1. ‘Selective estrogen receptor modulators (SERMs), tamoxifen and raloxifene, are commonly used in the treatment of BC, osteoporosis, and postmenopausal symptoms and are the only two FDA approved compounds for primary prevention of BC [4, 32].’ -What about aromatase inhibitors? Exemestane in the USA?? Anastrozole has just been approved by the MHRA in UK. Also SERMS are NOT used for post menopausal symptoms as they make them worse!

We had a section that touched on the use of AI in a prevention setting. We were aware that FDA has not approved AI for prevention but can be use off-labeled for this. However, we were not aware of MHRA approval for AI in the UK. We thank the reviewer for pointing this out. We have updated this section to mention this regulatory approval.

  1. ‘Additional studies of tamoxifen for *preventive agent reported these adverse effects as major factors contributing to discontinuation of treatment, especially among women taking tamoxifen for primary prevention as opposed to adjuvant therapy for BC [41, 42].’ -the section on tamoxifen is a joke. There is much more important long term data from the IBIS trials that is completely ignored. NASBP closed after only 4 years follow up and has no useful long term data

We have made this correction.  Updated references that included long term results and added this to the text as well. We have now included reference to IBIS trial: https://www.thelancet.com/journals/lanonc/article/PIIS1470-2045%2814%2971171-4/fulltext

  1. ‘Direct comparison of tamoxifen and raloxifene revealed equal risk reduction of invasive BC, but endometrial cancer in postmenopausal women, thromboembolic events, and stroke occurred in both groups [45].’-What about longer term follow up? -this showed tamoxifen was better for risk reduction?

     The STAR trials shutdown in 2012, the latest update was in 2010, these are incorporated into our review.  This means that we cannot add any additional long-term data specifically with this trials because the longest follow up the trial has is ~6 years before the entire study concluded. This trial will not be providing more long-term follow ups.

Round 2

Reviewer 3 Report

Comments and Suggestions for Authors

The authors have attempted to modify the introduction based on my many concerns. They insist on pushing back on a number of issues. I’m afraid they demonstrate they are not the experts here. Yet they show a lot of bias against the standard currently available risk reduction methods. They MUST dial down this bias and again rewrite the introduction using all the available literature not just the literature that backs up their biases. The remainder of the review is reasonable as it goes through other potential strategies in investigation

Specific points

1.       Abstract: ‘One in eight women will develop breast cancer in the US. For women with moderate (15-20%) to average (12.5%) risk of breast cancer, there are few options available for risk reduction. For high-risk (>20%) women’ _I do not recognise this classification. In Europe high risk is 30% or higher. Please justify this classification with a proper guideline

2.       Abstract ‘Despite their effectiveness in risk reduction, not many high-risk individuals opt for surgical (<50%) or hormonal (<10%) interventions due to severe side effects and potentially life-changing outcomes as key deterrents.’ -This language again shows inherent bias. The authors asked for a reference to higher uptake https://pubmed.ncbi.nlm.nih.gov/37005486/. Other reviews also show that uptake can be higher than 10% https://pubmed.ncbi.nlm.nih.gov/24594998/

3.       ‘. Non-modifiable risk factors that increase cumulative exposure to estrogen include older age at natural menarche and menopause. [3].’ -NO its younger age at menarche and older age at menopause!

4.       ‘prophylactic dual mastectomy [6, 7].’ -what is ‘dual’ doing here? The authors insist on using ‘prophylactic’ which is not much used now and  I disagree with their assertions about using risk reducing. I have never heard the term ‘dual’ in this context before

5.       ‘(Dual prophylactic mastectomy.) Dual prophylactic mastectomy is a highly invasive procedure that can be executed as a total mastectomy’ -Delete one of these

6.       ‘Genetic counseling based on 20-gene panels captures the most frequent mutations [6, 7, 9, 10]’ Ref 7 is not a publication on gene testing but PRS testing please delete. Ref 6 states ‘Women were tested with a 25-gene hereditary cancer panel including BRCA1/2 and 7 additional genes known to be associated with a >20% lifetime risk for breast cancer (ATM, CHEK2, PALB2, TP53, PTEN, CDH1, and STK11)’ So 9 genes only. Where do the authors get 20 genes from? Please justify

7.       ‘My query 10. I used the term ‘actionable’ which is a two-fold risk. There were Not 16 genes from BRIDGES ‘The BRIDGES study find evidence of elevated risk with 15 genes (https://www.nejm.org/doi/full/10.1056/nejmoa1913948) ‘

8.       My Query. ‘Women with other genetic predispositions including (mutation) in CDH1…’ -Most journals now use pathogenic variants’

Author response: ‘Pathogenic variants imply, at least to us, causality. There are some variants of unknown significance. Without functional demonstration, we find mutation a more neutral term since not all DNA variants will have an impact on risk (e.g, mutation in intron, enhancer, promoter). However, we are open to changing this if reviewer finds this adds value to the review.’ -This response is nonsensical. You are using the term in conjunction with the ‘word’ predisposition’ which strongly implies causality. Please use the ‘correct’ term

9.       The authors are clearly heavily biased against current standard risk reducing strategies. They MUST give a balanced view based on ALL the literature not just cherry picking the lit that support their bias. This small publication should not be given as the sole evidence ‘Unfortunately, 70% of patients suffer from pain or breast discomfort and 75-85% had decreased sexual interest; quality of life and surgery regret was negligible [27]’ Other much larger series show much better outcomes in bilateral risk reducing mastectomy that are better with no cancer diagnosis https://pubmed.ncbi.nlm.nih.gov/34219040/

Comments on the Quality of English Language

OK

Author Response

Dear Editorial Team,

We appreciate the overall positive comments and constructive critiques provided by Reviewer #3. We address reviewer’s concerns point by point below and we have accordingly made revisions throughout the main text. Reviewers #1 and #2 were satisfied with the revised manuscript and there is no longer an editorial concern with language usage.

The new revisions are indicated with markup tracked changes. We hope that these revisions and additional information provided are considered responsive to Reviewer #3’s remaining concerns and satisfy both Reviewer #3 and the editorial team.

The authors have attempted to modify the introduction based on my many concerns. They insist on pushing back on a number of issues. I’m afraid they demonstrate they are not the experts here. Yet they show a lot of bias against the standard currently available risk reduction methods. They MUST dial down this bias and again rewrite the introduction using all the available literature not just the literature that backs up their biases. The remainder of the review is reasonable as it goes through other potential strategies in investigation.

We appreciate feedback from the reviewer and provided references to more comprehensively review the field. We agree with the reviewer that the main focus on this review is to discuss new investigational approach for primary prevention of breast cancer. It is not our intention to misrepresent current risk-reducing interventions. We clearly state that prophylactic (or risk-reducing mastectomy) and anti-estrogen interventions are effective at risk-reduction. Statements we made on the introduction are not our opinion but citations from experts in their field that may pertain more specifically to the US and risk assessment and management in the US. There may be differences between US, UK, Europe and other countries as how patients perceive and understand their risk, follow/comply with physician’s recommendation and trust physician’s recommendation, and perceive the impact of these interventions in their quality of life.  While we focus on US women with breast cancer or at risk of developing breast cancer, we added some notes and references to these potential differences between US, UK and other countries.

Specific points

  1. Abstract: ‘One in eight women will develop breast cancer in the US. For women with moderate (15-20%) to average (12.5%) risk of breast cancer, there are few options available for risk reduction. For high-risk (>20%) women’ _I do not recognise this classification. In Europe high risk is 30% or higher. Please justify this classification with a proper guideline

We are based in the US. We state in the abstract and introduction that there numbers we show are for women in the US. We also mainly focus on clinical trials and regulatory approval (FDA) in the US. We now add reference to risk stratification in UK/Europe for comparison with US. We added in the previous revision note on regulatory approval for AI for prevention in the UK as recommended by the reviewer (FDA has not approved this use yet).

US guidelines may not have as clear cut definition for moderate and high risk (https://emedicine.medscape.com/article/2247407-overview#:~:text=American%20Society%20of%20Clinical%20Oncology,-Updated%20ASCO%20practice&text=For%20premenopausal%20with%20increased%20risk,(ER)%E2%80%93positive%20breast%20cancer). ACS guidelines use 15% as cut off for elevated risk. FDA uses 20% as high risk for anti-estrogen intervention. Other agencies indicate 20-25% is considered high-risk.

We agree with reviewer that in Europe and in many studies 25% or 30% (e.g., using a RR >2) rather than 20% is the threshold for high-risk consideration. For the purpose of this review, we don’t think it is critical to start at 20% or 30%. It may slightly affect what would be considered a moderate- or high-risk individual but we are not making a big emphasis on this. The only point we are trying to make is that a woman with a risk of 20% (whether is moderate or at the cusp of high-risk) is a more difficult case to manage than a woman with a risk of >50%. 

We provide on the revised version a 2023 publication by Michaels et al. (https://pubmed.ncbi.nlm.nih.gov/36759097/), with the definition for average, moderate and high-risk individuals (>20%) that aligns with US guidelines from ACS, ASCO and NCCN.

We add now this to the introduction: “For women in the US, a 1.67% increased risk over 5 years or 20% increased risk over a 20-year period is considered a high-risk individual [3]. This review is focused on federal agency guidelines and regulatory approvals that pertain specifically to US women. It is worth noting that some of these guidelines are different in other countries. For example, for women in Europe, high-risk is defined as a > 30% lifetime risk [6]. This difference and other considerations may affect how risk-reducing interventions are applied, perceived, and complied with in different geographical regions and countries.”

  1. Abstract ‘Despite their effectiveness in risk reduction, not many high-risk individuals opt for surgical (<50%) or hormonal (<10%) interventions due to severe side effects and potentially life-changing outcomes as key deterrents.’ -This language again shows inherent bias. The authors asked for a reference to higher uptake https://pubmed.ncbi.nlm.nih.gov/37005486/. Other reviews also show that uptake can be higher than 10% https://pubmed.ncbi.nlm.nih.gov/24594998/

We appreciate these references. We have incorporated these to the review and accordingly change uptake for hormonal intervention can be higher (>20-40%) in the UK and other European countries.

Please note that the reference you provide mostly deal with UK or other European women. Reference from 2023 and reference 31 within this article point to less than 8% uptake of hormonal interventions in the US. Because both the abstract and the introduction are meant to reflect US women data and recommendation from US agencies such as the FDA. We propose to keep the <10% number but clearly qualified this is for US women and add your references to indicate that uptake can be higher in the UK/Europe, especially when women are provided ample information and resources to understand their risk, treatment options and anticipated side effects and other considerations.

In the provided 2014 reference https://pubmed.ncbi.nlm.nih.gov/24594998/ the reported uptake is 11.6% (129 out of 1109). Table 1 in the discussion is useful as shows that in clinical trials uptake was up to 27% (35 out of 158). In non-clinical trial setting the highest uptake was 42% (57 out of 137) in Touch et al. 2004. This a US single institution study and authors do not report if women stop taking treatment after they agreed to this intervention.

In provided 2023 publication (https://pubmed.ncbi.nlm.nih.gov/37005486/) the authors state “While overall uptake of risk appointments was also lower than expected with only just over 40% of those at high risk taking up an appointment, the uptake of BCPM amongst attendees was exceptionally high at 77.5%. This is markedly higher than the average 10–11% uptake in our FHRPC utilising the same clinicians [29, 30], in other high risk population settings [29] and in eligible women in the USA at <8% [31].” So their 8% reference to the US is very similar to our <10% uptake in the US. Moreover, the authors recognize this is an exceptional high uptake and 60% of eligible women decline to participate so the 77.5% uptake in the 40% highly proactive women may not be applicable to women in the US or other contexts.

Again, it is beyond the scope of our review but it seems that in the US, women that adhere to tamoxifen treatment is lower than initial uptake and women in more recent years are more reluctant to take tamoxifen as reviewed in Padamsee et al 2017 and Jahan et al. 2021 (new reference, this if ref 31 from 2023 publication provided by the reviewer).

We have revised the text accordingly:

“For high-risk women in the US, FDA-approved primary prevention strategies include surgical removal of the breasts and/or ovaries, and the use of anti-estrogen therapies. Bilateral prophylactic mastectomy is currently the most effective procedure for preventing BC: it can reduce the incidence of BC by up to 90% in high-risk individuals [16]. Anti-estrogen treatments have been shown to reduce BC risk by up to 50% in high-risk women [16]. Though effective in risk reduction, less than 50% and less than 10% of high-risk individuals opt for surgical or hormonal interventions, respectively; with life-changing consequences and severe side effects as major contributing dissuading factors [16-18]. These prevention interventions are readily available, but may be underused due to: lack of clinician or patient information regarding risk level, lack of clinical confidence to discuss appropriate prevention options, personal social dynamics and fully informed choice, which can result in low uptake of prevention methods [16]. Bilateral prophylactic mastectomy uptake is well documented in women who are BRCA mutation carriers. However, on average only 20% of women at high-risk without the BRCA mutations undergo this surgical procedure but have reported ranges from 11-50%  [19, 20]. Population studies on hormonal interventions have reported low uptake for eligible women (1-5%), however, this falls short when compared to high-risk proactive women interested in using these interventions which can be as high as 40% [16, 21]. A recent study conducted in Europe showed that women who were informed of their risk and provided information on preventive options within 8 weeks of risk identification had a large increase in uptake (77.5%) of hormonal interventions compared to much lower uptake in standard clinical settings (11-20)% [18, 22-24]. Therefore, prevention interventions, either currently approved or investigational, should take into consideration education and informed decision-making in addition to clinically established risk reduction and management of adverse side effects.”

  1. ‘. Non-modifiable risk factors that increase cumulative exposure to estrogen include older age at natural menarche and menopause. [3].’ -NO its younger age at menarche and older age at menopause!

Yes, we apologize for this mistake. We restructured this sentence to address critique that menopause can be surgically induced and not just natural, and dropped the younger age at menarche.

The sentence now reads: “Non-modifiable risk factors that increase cumulative exposure to estrogen include younger age at  menarche and older age at natural menopause [2].”

  1. ‘prophylactic dual mastectomy [6, 7].’ -what is ‘dual’ doing here? The authors insist on using ‘prophylactic’ which is not much used now and  I disagree with their assertions about using risk reducing. I have never heard the term ‘dual’ in this context before

Yes, we apologize for this mistake. We meant to say double not dual. We are using now consistently “bilateral prophylactic mastectomy”.

Prophylactic mastectomy and risk-reducing mastectomy are synonymous. There are 78 pubmed entries in 2023 using “prophylactic mastectomy” (https://pubmed.ncbi.nlm.nih.gov/?term=%22prophylactic+mastectomy%22+2023&sort=pubdate&size=200 ) and 33 using “risk-reducing mastectomy” (https://pubmed.ncbi.nlm.nih.gov/?term=%22risk-reducing+mastectomy%22+2023&sort=pubdate&size=200 )

Another article (Cancers | Free Full-Text | Comparing Prognosis for BRCA1, BRCA2, and Non-BRCA Breast Cancer (mdpi.com) in this Special Issue of Cancers use the terms interchangeably. There are 10 mentions in the text to “prophylactic mastectomy” and 3 to “risk-reducing mastectomy”. This article uses more frequently risk-reducing surgeries than risk-reducing mastectomy. Thanks to reviewer’s input, we are using more broadly thoroughly at the text the terms “risk-reducing interventions” and “risk-reducing surgery”.

We propose to use prophylactic mastectomy instead of risk-reducing mastectomy to avoid sentences like these:

Risk-reducing mastectomy is a highly invasive procedure that can be executed as (..). In removing the entirety of the breast, this risk-reducing surgery removes the epithelial cells, the intended target cells, from which breast carcinomas arise”.

Risk-reducing mastectomy is an effective risk-reducing surgery”

  1. ‘(Dual prophylactic mastectomy.) Dual prophylactic mastectomy is a highly invasive procedure that can be executed as a total mastectomy’ -Delete one of these

      We have removed the double use of “Bilateral prophylactic mastectomy” which is updated with the new terminology.

The sentence now reads: “Bilateral prophylactic mastectomy.  This procedure completely removes the breast tissue in both breasts, including the ductal tree and surrounding stroma. Bilateral prophylactic mastectomy is a highly invasive procedure that can be executed as a total mastectomy, skin-sparing mastectomy, or total skin-sparing mastectomy.”

  1. ‘Genetic counseling based on 20-gene panels captures the most frequent mutations [6, 7, 9, 10]’ Ref 7 is not a publication on gene testing but PRS testing please delete. Ref 6 states ‘Women were tested with a 25-gene hereditary cancer panel including BRCA1/2 and 7 additional genes known to be associated with a >20% lifetime risk for breast cancer (ATM, CHEK2, PALB2, TP53, PTEN, CDH1, and STK11)’ So 9 genes only. Where do the authors get 20 genes from? Please justify

This is the 25 gene panel they used as in Table 1. They set risk at >20% life-time to consider it a pathogenic variant.

Breast cancer risk genes with certain risk estimates and currently with professional society guidelines for management

BRCA1, BRCA2,  CDH1  ,  PTEN,  STK11, TP53,  ATM,  CHEK2, PALB2      

 Breast cancer risk genes with uncertain risk estimates and currently without professional society guidelines for management

 BARD1, NBN

Genes for which there are currently no established breast cancer risks APC, BMPR1A, BRIP1, CDK4, CDKN2A, EPCAM, MLH1, MSH2, MSH6, MUTYH, PMS2, RAD51C, RAD51D, SMAD4

This is the 34 predisposition gene panel from BRIDGES study in Table 2. They set the risk at RR >2 to consider it a pathogenic variant.

ABRAXAS1,AKT1,ATM,BABAM2,BARD1,BRCA1,BRCA2,BRIP1,CDH1,CHEK2,EPCAM,FANCC,FANCM,GEN1,MEN1,MLH1,MRE11,MSH2,MSH6,MUTYH,NBN,NF1,PALB2,PIK3CA,PMS2,PTEN,RAD50,RAD51C,RAD51D,RECQL,RINT1,STK11,TP53,XRCC2

Yes, we agree that these studies show that there are 9-15 genes that are more informative to assess risk at the population level. We have modified the sentence to remove the reference to 25 or more susceptibility genes since not all of these are generally applicable. We have removed ref 7 from that sentence as recommended by the reviewer.

“Genetic counseling based on multi-gene panels that capture most frequent mutations [6, 9, 10]”

  1. ‘My query 10. I used the term ‘actionable’ which is a two-fold risk. There were Not 16 genes from BRIDGES ‘The BRIDGES study find evidence of elevated risk with 15 genes (https://www.nejm.org/doi/full/10.1056/nejmoa1913948) ‘

Yes, thanks for clarification. We are not sure what we need to do with this entry. BRIDGES study does not use the term “actionable” in the article. They set their classification of pathogenicity at RR >2. They use the term “informative”.

In the discussion they state that “Of the 10 genes that were regarded as having definitive evidence of an association with breast cancer risk in that analysis,10 7 genes had variants associated with breast cancer risk in our analysis; deleterious variants in the remaining 3 genes (CDH1PTEN, and STK11) are very rare and confer a predisposition to specific cancer syndromes.” We interpret this to mean that these predisposition genes have been linked to breast cancer in other studies in morr restricted family or lineage were other co-segregating genes or other factors contribute to risk. The low representation of variants (mutated alleles) of these predisposition genes in this study could not find a link in this population. This study uses only protein-truncating variants and nonsense variants, perhaps other variants (non-synonymous mutation that affect protein activity, promoter, enhancer, intron that affect gene expression could contribute to risk.

  1. My Query. ‘Women with other genetic predispositions including (mutation) in CDH1…’ -Most journals now use pathogenic variants’

Author response: ‘Pathogenic variants imply, at least to us, causality. There are some variants of unknown significance. Without functional demonstration, we find mutation a more neutral term since not all DNA variants will have an impact on risk (e.g, mutation in intron, enhancer, promoter). However, we are open to changing this if reviewer finds this adds value to the review.’ -This response is nonsensical. You are using the term in conjunction with the ‘word’ predisposition’ which strongly implies causality. Please use the ‘correct’ term

We think the difference is between the gene and the allele. As per BRIDGES studies and Rosenthal et al 2017 (ref 6), there is a list of predisposition genes that have been linked to breast cancer. A particular mutation or pathogenic variant (allele) of that gene was linked to increased breast cancer risk in at least a study. In other studies that gene is considered a genetic predisposition but not all mutations or variants of that gene (represented alleles) would be pathogenic – either at a >20% lifetime risk or RR >2.

In this context, we think that “mutation” is appropriate. In the case of BRCA mutation carriers we qualified that these individuals have >50% risk. We could say here BRCA pathogenic mutation carrier to mean the same thing. However, for  “genetic predispositions including mutations in ATM and CHK2, or who carry risk-associated alleles for multiple of the 92 susceptibility genes are generally considered moderate-risk individuals [8]”. Pathogenic variant define at >20% risk or RR >2 would not be appropriate in the context of moderate risk as we define at 15-20% as per Michaels et al. 2023.

  1. The authors are clearly heavily biased against current standard risk reducing strategies. They MUST give a balanced view based on ALL the literature not just cherry picking the lit that support their bias. This small publication should not be given as the sole evidence ‘Unfortunately, 70% of patients suffer from pain or breast discomfort and 75-85% had decreased sexual interest; quality of life and surgery regret was negligible [27]’ Other much larger series show much better outcomes in bilateral risk reducing mastectomy that are better with no cancer diagnosis https://pubmed.ncbi.nlm.nih.gov/34219040/

We thank the reviewer for this reference. This is 128 women study in the UK. We had a study with 98 women in Sweden, we add another reference from US women to have a more broader view on this point. We have added the following paragraph to address other reports.

“Bilateral prophylactic mastectomy studies focusing on patient-reported outcomes after breast reconstruction have noted higher body pain or breast discomfort and decreased sexual interest, but overall satisfaction with the procedure lowered cancer-related anxiety, and increased satisfaction with breast cosmesis [35-40]. However, it is important to note that these reports vary due to factors such as number of patients, the timing of reconstructive surgery, and pre- vs post-operation comparisons [35]. Nevertheless, the potential positive and negative impacts should be equally addressed with women for them to make fully informed decisions.”

Round 3

Reviewer 3 Report

Comments and Suggestions for Authors

The authors have now modified the introduction and this is much improved

I only have one comment on the US guidelines. 1.67% 5-year risk is the average risk for women over 60 years of age. These women are NOT high risk unless you are just using age alone

We add now this to the introduction: “For women in the US, a 1.67% increased risk over 5 years or 20% increased risk over a 20-year period is considered a high-risk individual [3]. 

Author Response

Dear Editorial Team,

We appreciate the overall positive comments and constructive critiques provided by Reviewer #3. We address reviewer’s concerns point by point below and we have accordingly made revisions throughout the main text.

The new revisions are indicated below. The second round of revisions addressed most of the concerns about the introduction section. We hope that these revisions and additional information provided are considered responsive to Reviewer #3’s remaining concern, and satisfy both Reviewer #3 and the Editorial team.

We also addressed concerns for Academic editor.

Academic Editor Response:

  • Reviewer 3 suggested to rephrase (introduction section) "all women have a risk " not just  "one in eight women" as written by authors.

We changed to this. We are not sure if this is the intended meaning the reviewer seeks: “With the number of new diagnoses still on the rise, one in eight women will develop BC within their lifetime, but all women are at risk.”

  • Retinoids and Rexinoids (section): "approved FDA-approved" is it the repetition correct?

Yes, this is a repeat. The sentence now reads: “It is the only FDA-approved rexinoid for the treatment of cutaneous T-cell lymphoma.”

  • Prophylactic vaccines (section):"and metastatic BC patients in phase I clinical trials (NCT00807781) (89);and metastatic BC patients in phase I clinical trials (NCT00807781) (129)" seems to be unhelpful repetition.

Sentence now reads: “Using this DNA vaccination approach (Figure 3A), anti-mammaglobin–specific T cells are readily detected in treated preclinical models, metastatic BC patients in phase 1 clinical trials, and metastatic BC patients in phase 1 clinical trials (NCT00807781) [89, 129]. “

Reviewer 3 Responses:

The authors have now modified the introduction, and this is much improved.

We appreciate the reviewer’s thorough evaluations and references which have helped us to improve this manuscript.

I only have one comment on the US guidelines. 1.67% 5-year risk is the average risk for women over 60 years of age. These women are NOT high risk unless you are just using age alone.

Yes, we agree age is an important factor taken into account in these models. In the US, the average risk is 1.5% at 40, 2.5% at 50 and 3.5% at 60. We add now this to the introduction: “For women in the US, a 1.67% increased risk over 5 years at any age or 20% increased risk over a 20-year period is considered a high-risk individual [3]. 

https://www.maurerfoundation.org/age-breast-cancer-what-young-women-need-to-know/